# BLACK-BOX GRADIENT MATCHING FOR RELIABLE OFFLINE BLACK-BOX OPTIMIZATION

## ABSTRACT

Offline design optimization problem arises in numerous science and engineering applications including materials engineering, where expensive online experimentation necessitates the use of in silico surrogate functions to predict and maximize the target objective over candidate designs. Although these surrogates can be learned from offline data, their predictions are often inaccurate outside the offline data regime. This challenge raises a fundamental question about the impact of imperfect surrogate model on the performance gap between its optima and the true oracle optima, and to what extent the performance loss can be mitigated. Although prior work developed methods to improve the robustness of surrogate models and their associated optimization processes, a provably quantifiable relationship between an imperfect surrogate and the corresponding performance gap, and whether prior methods directly address it, remain elusive. To shed more light on this important question, we present a novel theoretical formulation to understand offline black-box optimization, by explicitly bounding the optimization quality based on how well the surrogate matches the latent gradient field that underlines the offline data. Inspired by our theoretical analysis, we propose a principled black-box gradient matching algorithm to create effective surrogate models for offline optimization. Experiments on diverse real-world benchmarks demonstrate improved optimization quality using our approach to create surrogates.

## 1 INTRODUCTION

Many science and engineering applications involve optimizing an *expensive-to-evaluate black-box objective function* over large design spaces. Some examples include design optimization over candidate molecules, proteins (Nguyen & Daugherty, 2005), drugs, biological sequences, and superconducting materials (Si et al., 2016). To evaluate candidate designs, we need to perform physical lab experiments or computational simulations which are labor-intensive and impractical to do in an online manner. *Offline optimization* (Trabucco et al., 2022; 2021) is a more practical setting where we assume the access to a dataset of input and objective function evaluation pairs, and the overall goal is to use this offline training data to uncover optimal designs from the given input space.

The prototypical approach (Hutter et al., 2011; Brookes et al., 2019) to solve offline optimization problems is to learn a surrogate model from the given training data which can predict the objective function value for unknown inputs and find optimal input (i.e., maximizer) for this surrogate using gradient-based methods. The key implicit assumption behind this approach is that we can learn an accurate surrogate model over the entire input space using supervised learning. However, this is rarely achievable in practice due to the size and sparsity of the offline training data. In most cases, the surrogate model is only reliable within a constrained neighborhood of the offline data (Fannjiang & Listgarten, 2020) and can be highly erroneous outside this neighborhood. Consequently, there will be a discrepancy between the gradient fields of the Oracle (i.e., true objective function) and the surrogate model which will misguide the gradient search towards sub-optimal solutions.

This raises two related fundamental questions. *First*, how does the discrepancy in gradient estimation affect the performance gap between the optima of the surrogate model and the Oracle. *Second*, how to learn surrogate models which can closely approximate the gradient field of the Oracle. Both questions are challenging given that the Oracle's gradient field is entirely non-observable even at the offline training data points, and have not been studied by prior work. In fact, we note that while

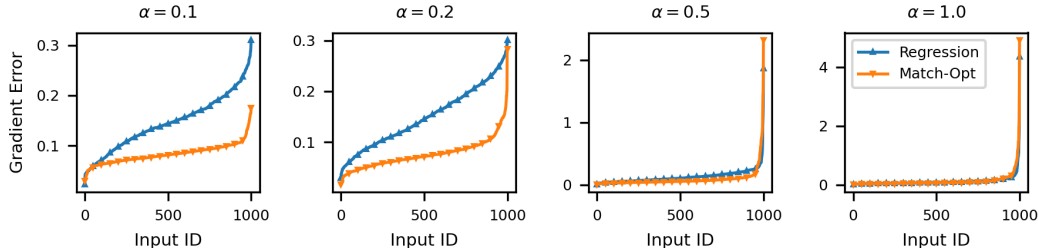

Figure 1: Comparison of gradient estimation error incurred by MATCH-OPT (orange) and standard regression (blue) while learning the gradient field of the Shekel function on 4-dimensional input space at different out-of-distribution (OOD) settings where test inputs were drawn from $\mathbb{N}(0, \alpha\mathbf{I})$ while training inputs were drawn from $\mathbb{N}(0, \mathbf{I})$. Smaller $\alpha$ indicates larger deviation from the offline data regime, which widens the performance gap between MATCH-OPT and standard regression.

there is an existing literature on random gradient estimation methods (Fu, 2015; Wang et al., 2018), those methods require the ability to actively sample data from the Oracle (or black-box function) which is not possible in the context of offline optimization.

**Contributions.** The main contribution of this paper is therefore to (1) provide theoretically-sound answers to these two questions; and (2) to demonstrate their practical significance on real-world offline design optimization problems:

**1.** To answer the first question, we present a theoretical framework to characterize the performance of gradient-based search guided by a surrogate model for offline optimization. We provably bound the performance gap between the optima of the Oracle and the trained surrogate as a function of how well the surrogate matches the (latent) gradient field of the Oracle on the offline training data. Our derived bound is non-trivial and yet model-agnostic, making it broadly applicable (Section 3).

**2.** To answer the second question, we present a principled gradient matching algorithm, referred to as MATCH-OPT, that is inspired by our theoretical analysis. Intuitively, gradients of the Oracle are relatively less sensitive to changes in the input. Hence, a surrogate model trained to directly match gradients will result in good offline optimization performance with gradient search from diverse starting points (referred as "reliable"). An overview of our algorithm is given in Fig. 2. Our algorithm MATCH-OPT is model-agnostic and allows us to approximate the gradient field that underlies the offline training data using a parametric surrogate (Section 4). In practice, existing offline optimization algorithms exhibits high variance in their performance across diverse design optimization tasks. MATCH-OPT is aimed at achieving reliable performance to address this critical challenge.

To provide an intuition and sanity check to readers, we visualize the reliability of our method's gradient estimation in several out-of-distribution (OOD) settings. We train our method, MATCH-OPT, and a standard regression model on the same set of random inputs drawn from $\mathbb{N}(0, \mathbf{I})$ and their Shekel function (https://www.sfu.ca/~ssurjano/shekel.html) evaluations. Fig. 1 plots the (sorted) gradient estimation error (i.e., the norm difference between predicted and oracle gradients) achieved by the two approaches at 1000 random inputs drawn from different OOD distributions $\mathbb{N}(0, \alpha\mathbf{I})$ parameterized with different values of $\alpha \in [0.1, 0.2, 0.5, 1.0]$.

It is observed that (1) when the test and train distributions are the same ($\alpha = 1.0$), the performance of the two approaches are the same; but (2) when $\alpha$ decreases (i.e., larger deviation from the offline data regime), our approach achieves significantly smaller error, suggesting that a direct gradient matching is more reliable in OOD data regimes. While this behaviour does not necessarily translate into better predictive accuracy, our Theorem 1 demonstrates that it will indeed minimize the optimization risk as we follow the surrogate gradient to find the oracle maximum. We note that similar ideas have shown great empirical success in a different area of structured prediction where models were learned to guide greedy search in combinatorial spaces (Doppa et al., 2014).

**3.** Finally, we demonstrate the efficacy of MATCH-OPT on diverse real-world optimization problems from the design-bench benchmark (Trabucco et al., 2022). Our results show that MATCH-OPT consistently shows improved optimization performance over existing baselines, and produces high-quality solutions with gradient search from diverse starting points (Section 5). Our code is provided in the supplementary files for review purposes and will be made public.

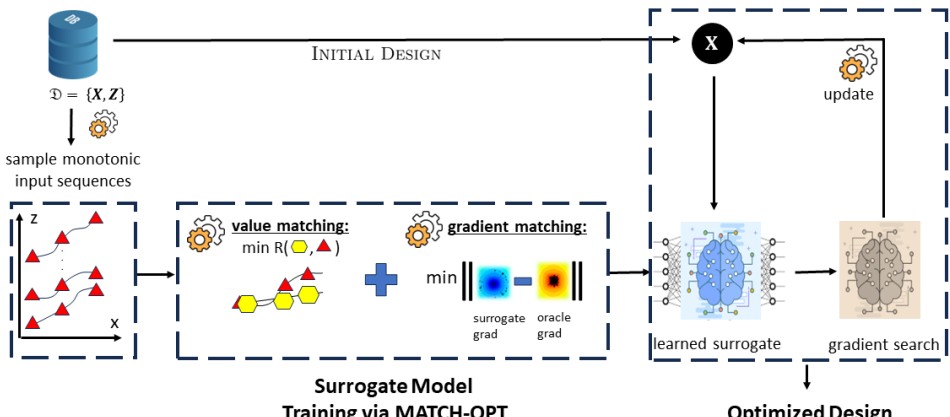

Figure 2: Our approach `MATCH-OPT` synthesizes input sequences with monotonically increasing function values from the offline dataset, which are used to train a parametric surrogate model. Our loss function incorporates both standard regression loss (i.e., value matching) and a novel gradient matching loss. We perform gradient search on the trained surrogate to find optimized designs.

## 2 BACKGROUND AND PROBLEM SETUP

**Offline Black-box Optimization.** Suppose $\mathfrak{X}$ is an input space where each $\mathbf{x} \in \mathfrak{X}$ is a candidate input. Let $g : \mathfrak{X} \mapsto \Re$ be an unknown, expensive real-valued objective function which can evaluate any given input $\mathbf{x} \in \mathfrak{X}$ to produce output $y = g(\mathbf{x})$. For example, in material design application, $g(\mathbf{x})$ corresponds to running a physical lab experiment. Our overall goal is to find an optimal input or design $\mathbf{x}_* \in \mathfrak{X}$ that maximizes the output of an experiment or simulation process $g(\mathbf{x})$,

$$\mathbf{x}_* \quad \triangleq \quad \arg\max_{\mathbf{x} \in \mathfrak{X}} \ g(\mathbf{x}) \ . \tag{1}$$

We are provided with a static dataset of $n$ input-output pairs $\mathfrak{D} = \{(\mathbf{x}_1, z_1), (\mathbf{x}_2, z_2), \cdots, (\mathbf{x}_n, z_n)\}$ collected *offline*, where $z_i = g(\mathbf{x}_i)$. The optimization algorithm does not have access to objective function $g$ values on inputs outside the dataset $\mathfrak{D}$.

**Surrogate Model.** We do not have access to the black-box function $g(\mathbf{x})$ beyond the offline dataset $\mathfrak{D}$ of $n$ training examples. This allows us to learn a surrogate $g_\phi(\mathbf{x})$ for $g(\mathbf{x})$ via supervised learning.

$$\phi \quad \triangleq \quad \arg\min_{\phi'} \sum_{i=1}^{n} \ell\left(g_{\phi'}\left(\mathbf{x}_i\right), g\left(\mathbf{x}_i\right)\right) \ = \ \arg\min_{\phi'} \sum_{i=1}^{n} \ell\left(g_{\phi'}\left(\mathbf{x}_i\right), z_i\right), \tag{2}$$

where $\phi$ denotes the parameters of surrogate model and $\ell(z, z')$ denotes the loss of predicting $z$ when the oracle value is $z'$ for a given input $\mathbf{x}$. For example, $\ell(z, z') = (z - z')^2$ and $g_\phi(\mathbf{x}) = \phi^\top \mathbf{x}$.

**Gradient-based Search Procedure.** Once learned, $\phi$ is fixed and we can use $g_\phi(\mathbf{x})$ as a surrogate to approximate the optimal design as:

$$\mathbf{x}_\phi \quad \simeq \quad \mathbf{x}_\phi^m \quad \text{where} \quad \mathbf{x}_\phi^{k+1} \triangleq \mathbf{x}_\phi^k \ + \ \lambda \cdot \nabla g_\phi\left(\mathbf{x}_\phi^k\right) \tag{3}$$

which is defined recursively for $0 \le k \le m-1$ via a $m$-step gradient ascent process starting from an initial solution $\mathbf{x}_\phi^0 = \mathbf{x}^0$ with a fixed learning rate $\lambda > 0$. The final iterate $\mathbf{x}_\phi^m$ is referred to as the solution of gradient search guided by the surrogate. Ideally, we want this solution to match the solution of gradient search guided by the oracle derivative, or the derivative of a differentiable proxy function that is closest to the ground-truth function $g$, if the oracle function does not exist. We refer to this as the oracle gradient, and the solution guided by the oracle gradient is defined as:

$$\mathbf{x}_* \quad \simeq \quad \mathbf{x}_*^m \quad \text{where} \quad \mathbf{x}_*^{k+1} \triangleq \mathbf{x}_*^k \ + \ \lambda \cdot \nabla g\left(\mathbf{x}_*^k\right) \tag{4}$$

which forms a similar gradient search of $m$ steps with the same initial solution $\mathbf{x}_*^0 = \mathbf{x}_\phi^0 = \mathbf{x}^0$ and learning rate $\lambda > 0$. However, this search process employs the oracle gradient instead of the

surrogate gradient. A discrepancy between Oracle gradients and surrogate gradients can result in a performance gap between the objective function values of $\mathbf{x}_*^m$ and $\mathbf{x}_\phi^m$.

This paper therefore studies two related questions in the context of gradient search guided by a trained surrogate model for offline optimization.

**Q1.** How does the discrepancy between Oracle and surrogate gradients impact the quality of uncovered solutions? This will be discussed in Section 3.

**Q2.** How to learn surrogate models that can closely approximate Oracle gradients using the offline training data $\mathfrak{D}$? Building on the result of Section 3, this will be discussed in Section 4.

## 3 THEORETICAL ANALYSIS

We provide a rigorous theoretical analysis to answer **Q1**. We derive an upper-bound for the performance gap between gradient search guided by the Oracle function and the trained surrogate, which is characterized explicitly in terms of how well the surrogate's gradient field fits with the offline data.

**Performance Gap.** First, we define the performance of the solution found via $m$ steps of gradient ascent on $g_\phi(\mathbf{x})$ starting from $\mathbf{x}_\phi^0 = \mathbf{x}^0$ via

$$\mathfrak{R}_{g_\phi}^m(\mathbf{x}_\phi^0) = \mathfrak{R}_{g_\phi}^m(\mathbf{x}^0) = g(\mathbf{x}_*) - g(\mathbf{x}_\phi) = g(\mathbf{x}_*) - g(\mathbf{x}_\phi^m). \tag{5}$$

where $\mathbf{x}_\phi^m$ is defined in equation 3. Similarly, we have $\mathfrak{R}_g^m(\mathbf{x}_*^0) = \mathfrak{R}_g^m(\mathbf{x}^0) = g(\mathbf{x}_*) - g(\mathbf{x}_*^m) \geq 0$. Note that we are distinguishing between the Oracle solution $\mathbf{x}_*^m$ and $\mathbf{x}_*$ here because often, finding $\mathbf{x}_*$ is intractable even with access to the Oracle $g(\mathbf{x})$ (e.g., combinatorial spaces). Thus, it is more practical to compare the surrogate solution with the oracle solution, rather than the oracle optima. We can now define the performance gap and state our main result.

**Definition 1.** *For a fixed gradient ascent process starting from $\mathbf{x}$ with $m$ update steps and learning rate $\lambda > 0$, the performance gap between the surrogate solution $\mathbf{x}_\phi^m$ and the oracle solution $\mathbf{x}_*^m$ is*

$$\mathfrak{G}_{m,\lambda}(\mathbf{x}) \triangleq \left\| \mathfrak{R}_g^m(\mathbf{x}) - \mathfrak{R}_{g_\phi}^m(\mathbf{x}) \right\|, \tag{6}$$

*where $\mathfrak{R}_g$ and $\mathfrak{R}_{g_\phi}$ are as defined above.*

**Theorem 1.** *Suppose $g(\mathbf{x})$ is a $\ell$-Lipschitz continuous and $\mu$-Lipschitz smooth function. The worst-case performance gap between $g$ and some arbitrary surrogate $g_\phi$ is upper-bounded by:*

$$\mathfrak{G}_{m,\lambda} \triangleq \max_{\mathbf{x}} \mathfrak{G}_{m,\lambda}\left(\mathbf{x}\right) \leq m\lambda\ell\left(1 + \lambda\mu\right)^{m-1} \cdot \max_{\mathbf{x}} \left\| \nabla g(\mathbf{x}) - \nabla g_\phi(\mathbf{x}) \right\|. \tag{7}$$

*Please see Appendix A for a detailed derivation and further discussion..*

Theorem 1 establishes that the worst-case performance gap between the surrogate and oracle solutions is upper-bounded by the maximum norm difference between the surrogate and oracle gradients over the input space. This provides a direct quantification of optimization quality as a function of gradient discrepancies. In addition, the result of Theorem 1 also characterizes a balance between the risk and potential of gradient search in terms of the learning rate and the number of update steps.

As the learning rate $\lambda$ or the number of search steps $m$ approaches zero, the bound in Theorem 1 also approaches zero. This means an extremely conservative gradient search (one that barely moves) would minimize the gap between $\mathfrak{R}_{g_\phi}$ and $\mathfrak{R}_g$, making the performance of the surrogate solution arbitrarily close to that of the oracle solution. At the same time, such a conservative strategy would widen the gap between the oracle solution and the optima, which ultimately deteriorates the overall performance of offline optimization. Conversely, an explorative search that uses larger $\lambda$ and $m$ will bring $\mathfrak{R}_g$ closer to zero, making the oracle solution arbitrarily close to the true optima. At the same time, it also widens the gap between the surrogate and oracle solution, again reducing the performance of offline optimization. Furthermore, as the bound in equation 7 holds for all possible choices of $g_\phi$, we can tighten it with respect to $g_\phi$. That is:

$$\mathfrak{G}_{m,\lambda} \leq m\lambda\ell\left(1 + \lambda\mu\right)^{m-1} \cdot \min_\phi \max_{\mathbf{x}} \left\| \nabla g(\mathbf{x}) - \nabla g_\phi(\mathbf{x}) \right\|. \tag{8}$$

For a fixed gradient based search configuration $(m, \lambda)$, the offline optimization task is therefore reduced to solving a minimax program,

$$\phi_* \quad = \quad \arg\min_{\phi} \max_{\mathbf{x}} \left\| \nabla g(\mathbf{x}) - \nabla g_\phi(\mathbf{x}) \right\| . \tag{9}$$

which is non-trivial since we do not have direct access to $\nabla g(\mathbf{x})$. Instead, we only have the value of $g(\mathbf{x}_i)$ at a finite number of inputs $\{\mathbf{x}_i\}_{i=1}^n$. We will describe the solution to equation 9 below.

---

**Algorithm 1** MATCH-OPT: Black-Box Gradient Matching from Offline Training Data

---

**Input**: Dataset $\mathfrak{D} = \{(\mathbf{x}_i, z_i)\}_{i=1}^n$, initial surrogate model parameters $\phi$, length of monotonic synthetic paths $m$, number of iterations $\tau$, learning rate $\lambda > 0$, regularization parameter $\alpha > 0$
**Output**: Surrogate model $g_\phi$ with parameters $\phi^{(\tau)}$
 1: Generate monotonic trajectories $\mathcal{C}^m$ via strategy from Krishnamoorthy et al. (2023b), Kumar et al. (2019)
 2: $\phi^{(0)} \leftarrow \phi$ // initialize parameters of surrogate model
 3: **for** $t \leftarrow 0 : \tau - 1$ **do**
 4: $\quad \mathfrak{L} \leftarrow 0$ // initialize the average loss
 5: $\quad$ **for** $\zeta = (\mathbf{x}_1, \ldots, \mathbf{x}_m) \in \mathcal{C}^m$ **do**
 6: $\quad\quad \mathfrak{L}_g^\zeta \leftarrow$ gradient matching loss using Eq. 12 with $\phi = \phi^{(t)}$
 7: $\quad\quad \mathfrak{L}_r^\zeta \leftarrow \alpha \cdot \sum_{i=1}^m \left( g(\mathbf{x}_i) - g_\phi(\mathbf{x}_i) \right)^2$ // compute regression regularizer with $\phi = \phi^{(t)}$
 8: $\quad\quad \mathfrak{L} \leftarrow \mathfrak{L} + \left| \mathcal{C}^m \right|^{-1} \left( \mathfrak{L}_g^\zeta + \mathfrak{L}_r^\zeta \right)$ // update the average loss
 9: $\quad \phi^{(t+1)} \leftarrow \phi^{(t)} + \lambda \cdot \nabla_\phi \mathfrak{L} \big|_{\phi = \phi^{(t)}}$ // once the inner loop finishes, $\mathfrak{L}$ in Eq. 13 will have been computed
10: **return** the learned surrogate model $g_\phi$ with $\phi = \phi^{(\tau)}$

---

## 4 PRACTICAL ALGORITHM: MATCH-OPT

This section answers **Q2** by providing a principled algorithm, which is referred to as MATCH-OPT. The crux of solving equation 9 lies with how we approximate the Oracle gradient field when we are given evaluations of the Oracle function at a fixed set of inputs (i.e., offline dataset). A naïve approach is to sample perturbed values around a target input and use the finite difference method to approximate its gradient (Fu, 2015; Wang et al., 2018). However, these methods require querying the Oracle function for perturbations of data points, which is not possible in the offline optimization setting. To overcome this challenge, we leverage the fundamental line integration theorem, which states that for any two inputs $\mathbf{x}$ and $\mathbf{x}'$ with corresponding values $z = g(\mathbf{x})$ and $z' = g(\mathbf{x}')$:

$$z - z' \;=\; g\Big(\mathbf{x}'\Big) - g\Big(\mathbf{x}\Big) \;\simeq\; \Big(\mathbf{x}' - \mathbf{x}\Big)^\top \int_0^1 \Big[ \nabla g_\phi\Big(\mathbf{x} \cdot (1-t) + \mathbf{x}' \cdot t\Big) \Big] \mathrm{d}t \,, \tag{10}$$

where the approximation holds when $\nabla g_\phi$ closely estimates the oracle gradient $\nabla g$. To enforce this, we need to find $\phi$ such that the averaged difference between the LHS and RHS of equation 10 is minimized. That is, we want to solve:

$$\phi^* \;=\; \arg\min_{\phi} \; \mathfrak{L}_g(\phi) \;\triangleq\; \mathbb{E}_{\mathbf{x}, \mathbf{x}' \in \mathfrak{D}} \left[ \left( \Delta z - \Delta \mathbf{x}^\top \int_0^1 \nabla g_\phi\Big(\mathbf{x} \cdot (1-t) + \mathbf{x}' \cdot t\Big) \mathrm{d}t \right)^2 \right] , \tag{11}$$

where $\Delta z = g(\mathbf{x}) - g(\mathbf{x}')$ provides a tractable learning objective when the expectation is taken over random inputs sampled from the offline training dataset $\mathfrak{D}$. We note that in the ideal scenario, equation 11 can be solved indirectly with a direct regression approach because the gradient fields of $g$ and $g_\phi$ must be the same when $g_\phi(\mathbf{x})$ accurately estimates $g(\mathbf{x})$ for every $\mathbf{x}$. However, as long as there are discrepancies, it is unclear which surrogate gradient (among surrogate candidates that approximate the Oracle equally well) would minimize the gradient discrepancy. As such, we argue that a direct gradient matching approach is more preferable in this case. This statement is supported by both our synthetic experiment (see Fig. 1) and real-world experiments presented in Section 5.3.

**Practical Considerations.** A naïve optimization of equation 11 requires enumerating over all pairs of training inputs. Iterating through the entire dataset is thus more expensive than a standard regression algorithm. To avoid this overhead, we adopt the strategy of Krishnamoorthy et al. (2023b) and

Kumar & Levine (2020), which organizes training data into monotonically increasing trajectories that mimic optimization paths. This encourages the model to learn the behavior of a gradient-based optimization algorithm, and thus allows the gradient matching algorithm to focus more on strategic input pairs that are more relevant for gradient estimation.

Specifically, let $\mathcal{C}^m$ denote a finite set of $m$-hop synthetic input paths with increasing objective function values, i.e., if $\zeta = \{\mathbf{x}_1, \mathbf{x}_2, \ldots, \mathbf{x}_m\} \in \mathcal{C}^m$, we have $g(\mathbf{x}_{i+1}) \geq g(\mathbf{x}_i)$. To sample trajectories from this set, we first bin the offline inputs based on their percentiles in the dataset, and subsequently sample one input from each bin to form a trajectory with monotonically increasing function values. We adapt the loss function in equation 11 to optimize along such sampled paths, and thus focus on estimating gradient information that is relevant to the downstream search procedure. That is, we aim to minimize $\mathfrak{L}_g(\phi; \mathcal{C}^m) \triangleq \mathbb{E}_{\zeta \in \mathcal{C}^m}[\mathfrak{L}_g(\phi; \zeta)]$, where:

$$
\begin{aligned}
\mathfrak{L}_g(\phi; \zeta) &\triangleq \sum_{i=1}^{m-1}\left(\Delta z - \Delta\mathbf{x}^\top \int_0^1 \nabla g_\phi\Big(\mathbf{x}_i \cdot (1-t) + \mathbf{x}_{i+1} \cdot t\Big)\mathrm{d}t\right)^2 \\
&\simeq \sum_{i=1}^{m-1}\left(\Delta z - \frac{1}{2\kappa}\sum_{u=1}^{\kappa}\left(\Delta\mathbf{x}^\top\Big(\nabla g_\phi\big(\mathbf{r}_i(u-1)\big) + \nabla g_\phi\big(\mathbf{r}_i(u)\big)\Big)\right)\right)^2,
\end{aligned}
\quad (12)
$$

and $\mathbf{r}_i(u) = \mathbf{x}_i \cdot (1 - (u/\kappa)) + \mathbf{x}_{i+1} \cdot (u/\kappa)$. Here, equation 12 takes empirical expectation over the successive pairs along the synthesized trajectories $\zeta \in \mathcal{C}^m$, $\Delta z \triangleq g(\mathbf{x}_{i+1}) - g(\mathbf{x}_i)$ and $\Delta\mathbf{x} \triangleq \mathbf{x}_{i+1} - \mathbf{x}_i$. In addition, the integral inside the expectation on the RHS of equation 12 is approximated via a discretization of $(0, 1)$ into $\kappa$ intervals with equal lengths. Our empirical inspections suggest that a discretization $\kappa = 5$ works best in practice. We combine this loss function with the regression loss along the synthetic trajectory to achieve the best of both worlds; that is:

$$
\mathfrak{L}(\phi) \triangleq \mathfrak{L}_{g,\mathcal{C}^m}(\phi) + \alpha \cdot \mathbb{E}_{\zeta \in \mathcal{C}^m}\left[\sum_{i=1}^{m}\left(g(\mathbf{x}_i) - g_\phi(\mathbf{x}_i)\right)^2\right],
\quad (13)
$$

where $\alpha$ is a trade-off hyper-parameter, and $\mathfrak{L}(\phi)$ denotes our ultimate loss function. A complete pseudo-code of this algorithm is detailed below (see Algorithm 1). We set $\alpha = 1$ in all our experiments since the regression and gradient match terms have the same unit scale.

**Complexity Analysis.** Given a $m$-hop synthetic sequence $\zeta$ of $d$-dimensional inputs, each step of the inner loop in Algorithm 1 will require a linear scan over $m$ segments. For each segment, the algorithm needs to compute (1) the gradient matching loss, which costs $\mathcal{O}(dm\kappa|\phi|)$ where $\kappa$ is the granularity of the discretization in equation 12 and $|\phi|$ is the number of parameters of the surrogate model, and (2) the regression regularizer on this path, which costs $\mathcal{O}(m|\phi|)$. Thus, suppose $p = |\mathcal{C}^m|$ synthetic input sequences/paths were generated for our algorithm, the entire inner loop of Algorithm 1 will incur a total cost of $\mathcal{O}(p \cdot (dm\kappa|\phi| + m|\phi|)) = \mathcal{O}(p \cdot dm\kappa|\phi|)$. This is the complexity per training iteration. For $\tau$ iterations, the total complexity of Algorithm 1 will be $\mathcal{O}(\tau \cdot p \cdot dm\kappa|\phi|)$.

## 5 EXPERIMENTS

This section describes the set of benchmark tasks used to evaluate and compare the performance of MATCH-OPT with those of other baselines (Section 5.1), the configurations of both our proposed algorithm and those baselines (Section 5.2), as well as their reported results (Section 5.3).

### 5.1 BENCHMARKS

Our empirical studies are conducted on 6 benchmark tasks from a diverse set engineering domains. Each task comprises a black-box oracle and an offline training dataset, which is a small subset of a much larger dataset used to train the oracle. Each participating algorithm only has access to the offline dataset. The oracle is only used to evaluate the performance of the final inputs recommended by those offline optimizers. The specifics of these datasets and their oracle functions are further provided in the design baseline package (Trabucco et al., 2022). Four tasks are defined over continuous input spaces, whereas the other two are discrete, which we summarize below.

**1 & 2.** The Ant Morphology (Brockman et al., 2016) (ANT) and D'Kitty Morphology dataset (Ahn et al., 2020) (DKITTY) collect morphological observations of two robots and their corresponding

rewards in moving as fast as possible, or towards a specific location. The morphological parameters of the robot is defined over a 60/56-dimensional continuous search space.

**3.** The Hopper Controller dataset (Ahn et al., 2020) (HOPPER) collects observations of a neural network policy weights and their rewards on the Hopper-v2 locomotion task in OpenAI Gym (Brockman et al., 2016). The search space is defined over 5126-dimensional continuous space.

**4.** The Superconductor dataset (Brookes et al., 2019) (SCON) collects observations of superconductor molecules and their corresponding critical temperatures. Each molecule is represented by a 86-dimensional continuous vector.

**5 & 6.** The TF-Bind-8 (TF8) and TF-Bind-10 (TF10) datasets (Barrera et al., 2016) collect the binding activity scores between a given human transcription factor and various DNA sequences of length 8 and 10 respectively. The goal of these *discrete* tasks is to find a DNA sequence that maximizes the binding score with the given transcription factor.

| METHOD | ANT | DKITTY | HOPPER | SCON | TF8 | TF10 | MNR |
|---|---|---|---|---|---|---|---|
| GA | 0.271 | 0.895 | 0.780 | 0.699 | 0.954 | 0.966 | 0.600 |
| ENS-MEAN | 0.517 | 0.899 | 1.524 | 0.716 | 0.926 | **0.968** | 0.500 |
| ENS-MIN | 0.536 | 0.908 | 1.42 | 0.734 | 0.959 | 0.959 | 0.467 |
| CMA-ES | **0.974** | 0.722 | 0.620 | **0.757** | **0.978** | 0.966 | 0.367 |
| MINS | 0.910 | 0.939 | 0.150 | 0.690 | 0.900 | 0.759 | 0.700 |
| CBAS | 0.842 | 0.879 | 0.150 | 0.659 | 0.916 | 0.928 | 0.733 |
| ROMA | 0.832 | 0.880 | 2.026 | 0.704 | 0.664 | 0.820 | 0.667 |
| BONET | 0.927 | 0.954 | 0.395 | 0.500 | 0.911 | 0.756 | 0.683 |
| COMS | 0.885 | 0.953 | **2.270** | 0.565 | 0.968 | 0.873 | 0.467 |
| **MATCH-OPT** | 0.931 (2) | **0.957 (1)** | 1.572 (3) | 0.732 (3) | 0.977 (2) | 0.924 (6) | **0.283** |

Table 1: Performance of MATCH-OPT and other baselines at $100^{th}$ percentile level. The last column shows the mean normalized rank (MNR) computed across all tasks (smaller is better). The individual rank of MATCH-OPT on each task is included next to its reported performance.

## 5.2 CONFIGURATION OF ALGORITHMS AND EVALUATION METHODOLOGY

**Baselines.** Our empirical studies evaluate and compare the performance of MATCH-OPT against those of multiple state-of-the-art baseline approaches including COMs (Trabucco et al., 2021), ROMA (Yu et al., 2021), BONET (Krishnamoorthy et al., 2023b). Several other baselines from the design bench benchmark (Trabucco et al., 2022) including Gradient Ascent (GA), Gradient Ascent Ensemble Mean (ENS-MEAN), Gradient Ascent Ensemble Min (ENS-MIN), covariance matrix adaptation evolution strategy (CMA-ES) (Hansen, 2006), model inversion networks (MINS) (Kumar & Levine, 2020), conditioning by adaptive sampling (CBAS) (Brookes et al., 2019) are also included for a thorough comparison. The same neural network architecture is used for all baselines. Details of our experiments are deferred to Appendix B.

**Evaluation Methodology.** Our experiments follow the widely adopted evaluation methodology introduced by Trabucco et al. (2022). That is, each algorithm starts the search from the same initial set of $n = 128$ offline inputs and generates the corresponding set of solution candidates which are evaluated by the oracle function. For each algorithm, these (128) solutions are then sorted in increasing order, and the corresponding values at the $100^{th}$ percentile (maximum solution) and $50^{th}$ (median solution) are reported in Table 1 and Table 2 below. All oracle values are normalized using the maximum and minimum values from a larger unobserved dataset (that was used to train the oracle). We run each algorithm on each task 4 times and report the mean and standard deviation. We report mean performance in the main text and defer their standard deviations to Appendix D.

**Comparison Metrics.** The overall performance of a baseline against other methods across different optimization tasks can be assessed using (a) their mean (normalized) performance; and (b) their mean (normalized) performance rank. While the first metric has often been used in prior work, it does not account for the variation in performance among tasks. For example, normalized performance are often close to 1 for easy tasks, whereas for harder tasks, they can be closer to 0. The mean performance metric therefore might favor algorithms that do well on easy tasks, but poorly on other hard tasks. To mitigate such biased assessment, we consider the mean normalized rank (MNR)

metric that is agnostic to such variation of performance. This is defined below:

$$\text{MNR}(\mathcal{A}) \triangleq \frac{1}{p} \sum_{i=1}^{p} \frac{\text{rank}(\mathcal{A}; \text{task}_i)}{\text{\# algorithms}} \tag{14}$$

where $p$ is the number of tasks and $\text{rank}(\mathcal{A}; \text{task}_i) = q$ means $\mathcal{A}$ is the $q$-best algorithm for the $i$-th task. To scale the MNR to the same range of $(0, 1)$ (for convenience), we also normalize the rank by the number of participating algorithms in the ranking order. An algorithm with low MNR therefore has more reliable performance across tasks, and is preferable to other methods with higher MNR.

## 5.3 RESULTS AND DISCUSSION

To demonstrate the effectiveness of MATCH-OPT, we report the $100^{\text{th}}$ and $50^{\text{th}}$ percentile results in Table 1 and Table 2 comparing MATCH-OPT with all the baselines. Other than the algorithm's individual performance reported for each task, we calculate its mean normalized rank (see equation 14) to account for the reliability of its performance (across tasks) in the comparison.

**Mean Rank Comparison.** Overall, no algorithm performs best in more than two task domains due to the diverse nature of the benchmark tasks. In fact, for the 100-percentile performance reported in Table 1, each algorithm only performs best in at most one task. Among these, MATCH-OPT performs best on the DKITTY dataset, and second best on ANT and TF8 datasets. MATCH-OPT is consistently among the top-3 performers on four out of six task domains, which is an evidence of its reliable performance. In fact, this is best reflected in terms of the mean normalized rank metric (MNR) which averages the normalized rank of each baseline across all six tasks (see equation 14). Among all participating algorithms, MATCH-OPT achieves the lowest MNR, which is also markedly lower than that of the second lowest MNR of COMS. At $50^{\text{th}}$ percentile, Table 2 also shows that MATCH-OPT achieves the best MNR among competing baselines.

| METHOD | ANT | DKITTY | HOPPER | SCON | TF8 | TF10 | MNR |
|---|---|---|---|---|---|---|---|
| GA | 0.130 | 0.742 | 0.089 | 0.641 | 0.510 | 0.794 | 0.600 |
| ENS-MEAN | 0.192 | 0.791 | 0.209 | 0.644 | 0.529 | **0.796** | 0.433 |
| ENS-MIN | 0.190 | 0.803 | 0.166 | **0.672** | 0.490 | 0.794 | 0.500 |
| CMA-ES | -0.049 | 0.482 | -0.033 | 0.590 | 0.592 | 0.786 | 0.683 |
| MINS | 0.614 | 0.889 | 0.088 | 0.414 | 0.420 | 0.465 | 0.650 |
| CBAS | 0.376 | 0.757 | 0.013 | 0.099 | 0.442 | 0.613 | 0.817 |
| ROMA | 0.448 | 0.760 | 0.370 | 0.420 | 0.560 | 0.780 | 0.533 |
| BONET | **0.620** | **0.897** | 0.390 | 0.470 | 0.505 | 0.465 | 0.417 |
| COMS | 0.557 | 0.879 | 0.379 | 0.414 | **0.652** | 0.606 | 0.467 |
| **MATCH-OPT** | 0.611 (3) | 0.887 (3) | **0.393 (1)** | 0.439 (6) | 0.594 (2) | 0.720 (6) | **0.350** |

Table 2: Performance of MATCH-OPT and other baselines at $50^{\text{th}}$ percentile level. The last column shows the mean normalized rank (MNR) computed across all tasks (smaller is better). The individual rank of MATCH-OPT on each task is included next to its reported performance.

**Reliability Assessment.** To further demonstrate the consistent reliability of MATCH-OPT as previously alluded to in the introduction section, we also plot the MNRs of all competing baselines at every solution percentile level in Fig. 3a. As expected, MATCH-OPT achieves the lowest MNR at almost every percentile, averaging at approximately $0.35$ which is again markedly lower than the second lowest MNR. In addition, we also plot the mean performance of the tested algorithms across all percentile level in Fig. 3b, which also show that MATCH-OPT is the best performer (on average) between 0- and 80-percentile. Above that, between 80- and 100-percentile level, MATCH-OPT is the second best performer. The above observations (both MNR and mean performance) suggest that MATCH-OPT is consistently the most reliable among all optimizers. We also refer the readers to Appendix E which further visualizes the entire rank distribution of the tested algorithm across different percentile level. All observations are consistent with our above observations in Fig. 3a.

## 6 RELATED WORK

Black-box optimization problems were previously approached using derivative-free methods, such as random gradient estimation (Wang et al., 2018) or Bayesian optimization (Snoek et al., 2012;

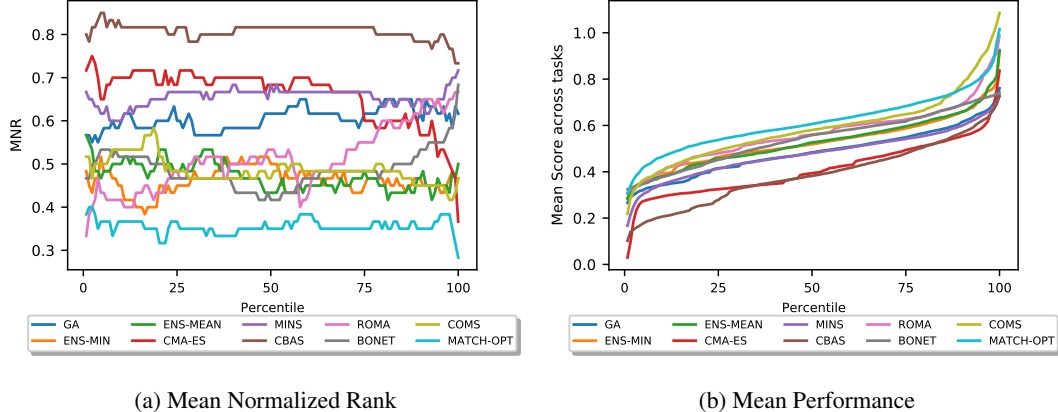

(a) Mean Normalized Rank                    (b) Mean Performance

Figure 3: Plots of (a) mean normalized ranks (MNRs); and (b) mean (normalized) performance of baselines at all performance percentile levels.

Wang et al., 2013; Eriksson et al., 2019). These methods require online evaluation of the oracle function to approximate its derivative or learn its surrogate model. In many practical applications, this can be very expensive (e.g., testing new protein or drug design), or even dangerous (e.g., test-driving autonomous vehicles in a real physical environment). To avoid this, *offline optimization* approaches tackle this problem via utilizing an existing dataset that records oracle evaluations for a fixed set of inputs. These approaches can be categorized into two main families:

**Conditioning Search Model.** Existing approaches in this direction are grounded in the framework of density estimation, which aims to learn a probabilistic prior over the input space. The search model is treated as a probability distribution conditioned on the rare event of achieving a high oracle score, and is estimated using different approaches, such as adaptive trust-region based strategies (Brookes et al., 2019), or zero-sum game (Fannjiang & Listgarten, 2020). **?** learns an inverse mapping of the oracle evaluations to inputs using conditional generative adversarial network (Mirza & Osindero, 2014) and uses it as a search model that predicts which regions will most likely have high-performing designs. These approaches often require learning a computationally expensive generative model of the input space, and are sensitive to the accuracy of the conditioning at out-of-distribution input regimes. The robustness of these conditioning algorithms has not been defined, nor investigated.

**Conditioning Surrogate Model.** Approaches in this direction tend to fix the search methodology and focus on conditioning the surrogate model to improve the likelihood of finding a good design. This is generally achieved via adopting different forms of regularization on the predicted values of OOD inputs based on the learned surrogate. For example, Yu et al. (2021) uses robust model pre-training and adaptation to ensure local smoothness, whereas Fu & Levine (2021) maximizes data likelihood to reduce the uncertainty in OOD prediction. Alternatively, Trabucco et al. (2021) penalizes high-value predictions for OOD examples to avoid overestimation of OOD inputs. These heuristic approaches are only justified empirically through practical demonstrations. From a theoretical perspective, the extent of effectiveness of these conditioning algorithms, as well as the fundamental question regarding when to trust a surrogate function both remain unclear.

## 7 CONCLUSION

This paper presents a new theoretical perspective on offline black-box optimization which established the first upper bound on the performance gap between the solutions guided by a trained surrogate and the oracle function. The bound reveals that such performance gap depends on how well the surrogate model matches the gradient field of the Oracle function on the offline dataset. Inspired by this theoretical analysis, we studied a novel algorithm for creating surrogate models based on gradient matching and demonstrated improved solutions on diverse real-world benchmarks. Although our theory and algorithm is grounded in the context of offline optimization, the developed principles can also be broadly applied to related sub-areas including safe Bayesian optimization and safe reinforcement learning in interactive online learning scenarios.

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

## A   PROOF OF THEOREM 1

**Theorem 1.** *Suppose $g(\mathbf{x})$ is a continuous function with Lipschitz and smooth constants, $\ell$ and $\mu$. Then, we have*

$$\mathfrak{G}_{m,\lambda} \triangleq \max_{\mathbf{x}} \mathfrak{G}_{m,\lambda}\left(\mathbf{x}\right) \leq m\lambda\ell\left(1+\lambda\mu\right)^{m-1} \cdot \max_{\mathbf{x}} \left\|\nabla g(\mathbf{x}) - \nabla g_{\phi}(\mathbf{x})\right\|$$

*which characterizes the upper-bound of the worst-case performance gap in terms of the maximum norm difference between the surrogate and oracle gradient over the input space.*

*Proof.* We first note that the performance of the $m$-step oracle solution starting at $\mathbf{x}_*^0 = \mathbf{x}^0$ is exactly the $(m\text{-}1)$-step oracle solution starting at $\mathbf{x}_*^1$. Likewise, the performance of the $m$-step surrogate solution starting at $\mathbf{x}_\phi^0 = \mathbf{x}^0$ is exactly the $(m\text{-}1)$-step surrogate solution starting at $\mathbf{x}_\phi^1$. That is:

$$\mathfrak{R}_g^m\left(\mathbf{x}^0\right) = \mathfrak{R}_g^m\left(\mathbf{x}_*^0\right) = \mathfrak{R}_g^{m-1}\left(\mathbf{x}_*^1\right) \quad \text{where} \quad \mathbf{x}_*^1 = \mathbf{x}_*^0 + \lambda\nabla g\left(\mathbf{x}_*^0\right)$$
$$\mathfrak{R}_g^m\left(\mathbf{x}^0\right) = \mathfrak{R}_{g_\phi}^m\left(\mathbf{x}_\phi^0\right) = \mathfrak{R}_{g_\phi}^{m-1}\left(\mathbf{x}_\phi^1\right) \quad \text{where} \quad \mathbf{x}_\phi^1 = \mathbf{x}_\phi^0 + \lambda\nabla g_\phi\left(\mathbf{x}_\phi^0\right) \qquad (15)$$

Consequently, for each initial point $\mathbf{x}^0$, we can bound the performance gap as follows:

$$\begin{aligned}
\mathfrak{G}_{m,\lambda}\left(\mathbf{x}^0\right) &\triangleq \left\|\mathfrak{R}_{g_\phi}^m\left(\mathbf{x}^0\right) - \mathfrak{R}_g^m\left(\mathbf{x}^0\right)\right\| = \left\|\mathfrak{R}_{g_\phi}^{m-1}\left(\mathbf{x}_\phi^1\right) - \mathfrak{R}_g^{m-1}\left(\mathbf{x}_*^1\right)\right\| \\
&= \left\|\mathfrak{R}_{g_\phi}^{m-1}\left(\mathbf{x}_\phi^1\right) - \mathfrak{R}_g^{m-1}\left(\mathbf{x}_\phi^1\right) + \mathfrak{R}_g^{m-1}\left(\mathbf{x}_\phi^1\right) - \mathfrak{R}_g^{m-1}\left(\mathbf{x}_*^1\right)\right\| \\
&\leq \left\|\mathfrak{R}_{g_\phi}^{m-1}\left(\mathbf{x}_\phi^1\right) - \mathfrak{R}_g^{m-1}\left(\mathbf{x}_\phi^1\right)\right\| + \left\|\mathfrak{R}_g^{m-1}\left(\mathbf{x}_\phi^1\right) - \mathfrak{R}_g^{m-1}\left(\mathbf{x}_*^1\right)\right\| \\
&= \mathfrak{G}_{m-1,\lambda}\left(\mathbf{x}_\phi^1\right) + \left\|\mathfrak{R}_g^{m-1}\left(\mathbf{x}_\phi^1\right) - \mathfrak{R}_g^{m-1}\left(\mathbf{x}_*^1\right)\right\|. \qquad (16)
\end{aligned}$$

Thus, let $\mathcal{E}_{m-1}(\mathbf{x}_\phi^1, \mathbf{x}_*^1) \triangleq \|\mathfrak{R}_g^{m-1}(\mathbf{x}_\phi^1) - \mathfrak{R}_g^{m-1}(\mathbf{x}_*^1)\|$, we have $\mathfrak{G}_{m,\lambda}(\mathbf{x}^0) \leq \mathfrak{G}_{m-1,\lambda}(\mathbf{x}_\phi^1) + \mathcal{E}_{m-1}(\mathbf{x}_\phi^1, \mathbf{x}_*^1)$. To bound the term $\mathcal{E}_{m-1}(\mathbf{x}_\phi^1, \mathbf{x}_*^1)$, we will prove the following intermediate results.

**Lemma 1.** *For any $k \in [1, m]$ and two different starting points $\mathbf{u}^0$ and $\mathbf{v}^0$, the performance gap between the $k$-step oracle solutions respectively starting from $\mathbf{u}^0$ and $\mathbf{v}^0$ is bounded by the norm distance between the starting points:*

$$\mathcal{E}_k\left(\mathbf{u}^0, \mathbf{v}^0\right) = \left\|\mathfrak{R}_g^k\left(\mathbf{u}^0\right) - \mathfrak{R}_g^k\left(\mathbf{v}^0\right)\right\| \leq \ell(1+\lambda\mu)^k\left\|\mathbf{u}^0 - \mathbf{v}^0\right\|. \qquad (17)$$

*Proof.* Let us first define the respective oracle search trajectories using the same gradient ascent formalism. That is, the respective candidate solutions at some intermediate step $\kappa \in [1, k]$ are given by $\mathbf{u}^\kappa = \mathbf{u}^{\kappa-1} + \lambda\nabla g(\mathbf{u}^{\kappa-1})$ and $\mathbf{v}^\kappa = \mathbf{v}^{\kappa-1} + \lambda\nabla g(\mathbf{v}^{\kappa-1})$. We can then make use of the Lipschitz continuous assumption to achieve the following bound:

$$\begin{aligned}
\left\|\mathfrak{R}_g^k\left(\mathbf{u}^0\right) - \mathfrak{R}_g^k\left(\mathbf{v}^0\right)\right\| &= \left\|g\left(\mathbf{x}_*\right) - g\left(\mathbf{u}^k\right) - g\left(\mathbf{x}_*\right) + g\left(\mathbf{v}^k\right)\right\| \\
&= \left\|g\left(\mathbf{v}^k\right) - g\left(\mathbf{u}^k\right)\right\| \leq \ell \cdot \left\|\mathbf{v}^k - \mathbf{u}^k\right\|. \qquad (18)
\end{aligned}$$

We subsequently bound the distance between the candidate solutions at step $k$ in terms of the distance at step $k-1$ using the smoothness conditions:

$$\begin{aligned}
\left\|\mathbf{v}^k - \mathbf{u}^k\right\| &= \left\|\mathbf{v}^{k-1} - \mathbf{u}^{k-1} + \lambda\nabla g\left(\mathbf{v}^{k-1}\right) - \lambda\nabla g\left(\mathbf{u}^{k-1}\right)\right\| \\
&\leq \left\|\mathbf{v}^{k-1} - \mathbf{u}^{k-1}\right\| + \lambda\left\|\nabla g\left(\mathbf{v}^{k-1}\right) - \nabla g\left(\mathbf{u}^{k-1}\right)\right\| \\
&\leq (1+\lambda\mu)\left\|\mathbf{v}^{k-1} - \mathbf{u}^{k-1}\right\|. \qquad (19)
\end{aligned}$$

Applying this bound recursively yields $\|\mathbf{v}^k - \mathbf{u}^k\| \leq (1+\lambda\mu)^k\|\mathbf{v}^0 - \mathbf{u}^0\|$. We finally substitute the above into equation 18 to arrive at the final bound $\|\mathfrak{R}_g^k(\mathbf{u}^0) - \mathfrak{R}_g^k(\mathbf{v}^0)\| \leq \ell(1+\lambda\mu)^k\|\mathbf{v}^0 - \mathbf{u}^0\|$. $\quad\square$

Applying Lemma 1 with $k = m - 1$, $\mathbf{u}^0 = \mathbf{x}_\phi^1$, and $\mathbf{v}^0 = \mathbf{x}_*^1$ subsequently allows us to bound $\mathcal{E}_{m-1}(\mathbf{x}_\phi^1, \mathbf{x}_*^1)$ as follows:

$$
\begin{aligned}
\mathcal{E}_{m-1}\left(\mathbf{x}_\phi^1, \mathbf{x}_*^1\right) &\leq \ell(1 + \lambda\mu)^{m-1} \left\| \mathbf{x}_\phi^1 - \mathbf{x}_*^1 \right\| \\
&= \ell(1 + \lambda\mu)^{m-1} \left\| \mathbf{x}^0 + \lambda\nabla g_\phi(\mathbf{x}^0) - \mathbf{x}^0 - \lambda\nabla g(\mathbf{x}^0) \right\| \\
&= \lambda\ell(1 + \lambda\mu)^{m-1} \left\| \nabla g_\phi(\mathbf{x}^0) - \nabla g(\mathbf{x}^0) \right\| \triangleq \mathcal{Q}\left(\mathbf{x}^0\right) .
\end{aligned}
\tag{20}
$$

We will now use this result to complete our bound for the performance gap. That is:

$$
\begin{aligned}
\mathfrak{G}_{m,\lambda} &\triangleq \max_{\mathbf{x}^0} \mathfrak{G}_{m,\lambda}\left(\mathbf{x}^0\right) \leq \max_{\mathbf{x}^0} \mathfrak{G}_{m-1,\lambda}\left(\mathbf{x}_\phi^1\right) + \max_{\mathbf{x}^0} \mathcal{Q}\left(\mathbf{x}^0\right) \\
&\leq \mathfrak{G}_{m-1,\lambda} + \max_{\mathbf{x}^0} \mathcal{Q}\left(\mathbf{x}^0\right) \\
&\leq \mathfrak{G}_{0,\lambda} + m \cdot \max_{\mathbf{x}^0} \mathcal{Q}\left(\mathbf{x}^0\right) ,
\end{aligned}
\tag{21}
$$

where the last inequality is obtained via recursively applying the previous inequality $m$ times. Substituting $\mathfrak{G}_{0,\lambda} = 0$ and the upper-bound for $\mathcal{Q}\left(\mathbf{x}^0\right)$ above into equation 21 gives:

$$
\mathfrak{G}_{m,\lambda} \leq m\lambda\ell(1 + \lambda\mu)^{m-1} \cdot \max_{\mathbf{x}^0} \left\| \nabla g_\phi\left(\mathbf{x}^0\right) - \nabla g\left(\mathbf{x}^0\right) \right\| ,
\tag{22}
$$

which completes our proof for Theorem 1. $\qquad\square$

**Tightness of the bound.** Note that despite the exponential dependence on $m$ of the above bound, its tightness can be controlled by choosing a sufficiently small value for $\lambda$. For example, if we choose $\lambda \leq 1/m$, it will follow that

$$
(1 + \lambda \cdot \mu)^{m-1} \leq \left(1 + \frac{\mu}{m}\right)^{m-1} < \left(1 + \frac{\mu}{m}\right)^m
\tag{23}
$$

which will approach $e^\mu$ in the limit of $m$. Here, we use the known fact that $\lim_{m\to\infty}(1 + \mu/m)^m = e^\mu$ with $\mu > 0$. As such, when $m$ is sufficiently large the bound in Theorem 1 is upper-bounded with $m \cdot \lambda \cdot \ell \cdot (1 + \lambda \cdot \mu)^{m-1} \cdot$ gradient-gap $\simeq \ell \cdot e^\mu \cdot$ gradient-gap which asserts that the worst-case performance gap of our offline optimizer is approaching (in the limit of $m$) $\ell \cdot e^\mu \cdot \max_{\mathbf{x}} \|\nabla g(\mathbf{x}) - \nabla g_\phi(\mathbf{x})\| = \mathbf{O}(\max_{\mathbf{x}} \|\nabla g(\mathbf{x}) - \nabla g_\phi(\mathbf{x})\|)$ which is not dependent on the number of gradient steps.

**Reducing the gradient gap.** Intuitively, minimizing Eq. 11 will reduce the gradient gap. To formalize this intuition rigorously, we will show below that (1) in the limit of optimization if a parameterization $\phi$ can be found that zeroes out the loss in Eq. 11 over the entire input space, the gradient gap is zero; and (2) in more practical cases, where the loss in Eq. 11 is not zero, the gradient gap is still guaranteed to be upper-bound by the Lipschitz constant of the function gap, which decreases as we optimize the loss function in Eq. 11. These are detailed below.

**A. The minimized loss in Eq. 11 is zero.** In this case, let us define:

$$
\mathbf{F}_\phi(\mathbf{x}, \mathbf{x}') \triangleq \int_0^1 \nabla g(t\mathbf{x} + (1-t)\mathbf{x}')\mathrm{d}t - \int_0^1 \nabla g_\phi(t\mathbf{x} + (1-t)\mathbf{x}')\mathrm{d}t
\tag{24}
$$

The loss in Eq. 11 can be rewritten as

$$
\mathfrak{L}_g(\phi) = \mathbb{E}\left[ \left( \mathbf{F}_\phi\left(\mathbf{x}, \mathbf{x}'\right)^\top \left(\mathbf{x} - \mathbf{x}'\right) \right)^2 \right]
\tag{25}
$$

where the expectation is over all pairs $(\mathbf{x}, \mathbf{x}')$ from the input space. At the optimal $\phi$, since $\mathfrak{L}_g(\phi) = 0$ as assumed, it follows that

$$
\mathbf{F}_\phi\left(\mathbf{x}, \mathbf{x}'\right)^\top \left(\mathbf{x} - \mathbf{x}'\right) = 0
\tag{26}
$$

for any choice of $(\mathbf{x}, \mathbf{x}')$. Next, by the line integration theorem, we also have

$$g(\mathbf{x}) - g(\mathbf{x}') = (\mathbf{x} - \mathbf{x}')^\top \left( \int_0^1 \nabla g(t\mathbf{x} + (1-t)\mathbf{x}') \mathrm{d}t \right) \tag{27}$$

$$g_\phi(\mathbf{x}) - g_\phi(\mathbf{x}') = (\mathbf{x} - \mathbf{x}')^\top \left( \int_0^1 \nabla g_\phi(t\mathbf{x} + (1-t)\mathbf{x}') \mathrm{d}t \right) \tag{28}$$

which together imply

$$\mathbf{F}_\phi\left(\mathbf{x}, \mathbf{x}'\right)^\top \left(\mathbf{x} - \mathbf{x}'\right) = g(\mathbf{x}) - g(\mathbf{x}') - g_\phi(\mathbf{x}) + g_\phi(\mathbf{x}') \tag{29}$$

Combining Eq. 26 and Eq. 29 results in

$$g(\mathbf{x}) - g_\phi(\mathbf{x}) = g(\mathbf{x}') - g_\phi(\mathbf{x}') \tag{30}$$

for any choice of $(\mathbf{x}, \mathbf{x}')$. This means there exists a constant $c$ such that

$$g(\mathbf{x}) - g_\phi(\mathbf{x}) = c \tag{31}$$

for all $\mathbf{x}$. Thus, taking the derivative with respect to $\mathbf{x}$ on both sides of the above yields

$$\nabla g(\mathbf{x}) - \nabla g_\phi(\mathbf{x}) = 0 \tag{32}$$

which implies immediately that the gradient gap is zero everywhere. Hence, optimizing Eq. 11 guarantees in principle that the gradient will be perfectly matched in the limit of data (i.e., when we take the expectation over the entire input space rather than over a finite set of offline data points).

**B. The minimized loss in Eq. 11 is not zero.** In this case, let us define

$$h(\mathbf{x}) = g(\mathbf{x}) - g_\phi(\mathbf{x}) \tag{33}$$

and it will follow that $|\mathbf{F}_\phi(\mathbf{x}, \mathbf{x}')^\top (\mathbf{x} - \mathbf{x}')| = |h(\mathbf{x}) - h(\mathbf{x}')|$ following Eq. 29 above. This means our loss function is working towards minimizing $(h(\mathbf{x}) - h(\mathbf{x}'))^2$ over $(\mathbf{x}, \mathbf{x}')$. This will makes $h(\mathbf{x})$ smoother as the output distance between different inputs are being reduced.

As a result, this process will reduce the Lipschitz constant $\epsilon$ of $h(\mathbf{x})$, which is defined to be the minimum value such that

$$|h(\mathbf{x}) - h(\mathbf{x}')| \leq \epsilon \cdot \|\mathbf{x} - \mathbf{x}'\| \tag{34}$$

which implies

$$|(g(\mathbf{x}) - g(\mathbf{x}')) - (g_\phi(\mathbf{x}) - g_\phi(\mathbf{x}'))| \leq \epsilon \cdot \|\mathbf{x} - \mathbf{x}'\| \tag{35}$$

Now, dividing both sides by $\|\mathbf{x} - \mathbf{x}'\|$ yields

$$\left| \frac{g(\mathbf{x}) - g(\mathbf{x}')}{\|\mathbf{x} - \mathbf{x}'\|} - \frac{g_\phi(\mathbf{x}) - g_\phi(\mathbf{x}')}{\|\mathbf{x} - \mathbf{x}'\|} \right| \leq \epsilon \tag{36}$$

Now, suppose we choose $\mathbf{x}' = \mathbf{x} + t \cdot \mathbf{e}_i$ where $\mathbf{e}_i$ is a $d$-dimensional one-hot vector with the hot component at the $i$-th position where $d$ denotes the input dimension. So, the above is equivalent to

$$\left| \frac{g(\mathbf{x} + t \cdot \mathbf{e}_i) - g(\mathbf{x})}{t\|\mathbf{e}_i\|} - \frac{g_\phi(\mathbf{x} + t \cdot \mathbf{e}_i) - g_\phi(\mathbf{x})}{t\|\mathbf{e}_i\|} \right| \leq \epsilon \tag{37}$$

or more expressively,

$$\epsilon \leq \frac{g(\mathbf{x} + t \cdot \mathbf{e}_i) - g(\mathbf{x})}{t\|\mathbf{e}_i\|} - \frac{g_\phi(\mathbf{x} + t \cdot \mathbf{e}_i) - g_\phi(\mathbf{x})}{t\|\mathbf{e}_i\|} \leq \epsilon \tag{38}$$

Taking $\lim_{t\to 0}$ on all parts of the above inequality, the above can be rewritten as

$$-\epsilon \leq \lim_{t\to 0} \left( \frac{g(\mathbf{x} + t \cdot \mathbf{e}_i) - g(\mathbf{x})}{t\|\mathbf{e}_i\|} \right) - \lim_{t\to 0} \left( \frac{g_\phi(\mathbf{x} + t \cdot \mathbf{e}_i) - g_\phi(\mathbf{x})}{t\|\mathbf{e}_i\|} \right) \leq \epsilon \tag{39}$$

Next, using the definition of directional gradient

$$\nabla_{\mathbf{r}} g(\mathbf{x}) \quad = \quad \lim_{t \to 0} \frac{1}{t} \left( g(\mathbf{x} + t \cdot \mathbf{r}) - g(\mathbf{x}) \right) \tag{40}$$

and the fact that $\nabla_{\mathbf{r}} g(\mathbf{x}) = \nabla g(\mathbf{x})^\top \mathbf{r}$ on $\mathbf{r} = \mathbf{e}_i$, we have

$$-\epsilon \quad \leq \quad \frac{1}{\|\mathbf{e}_i\|} \cdot \left( \nabla g(\mathbf{x})^\top \mathbf{e}_i \right) - \frac{1}{\|\mathbf{e}_i\|} \cdot \left( \nabla g_\phi(\mathbf{x})^\top \mathbf{e}_i \right) \quad \leq \quad \epsilon \tag{41}$$

which implies

$$\left( \nabla g(\mathbf{x}) - \nabla g_\phi(\mathbf{x}) \right)^\top \mathbf{e}_i \quad \leq \quad \epsilon \cdot \|\mathbf{e}_i\| \quad = \quad \epsilon \tag{42}$$

where the last step is true because $\|\mathbf{e}_i\| = 1$. Next, repeat the above argument with $\mathbf{r} = \mathbf{e}_i$ for $i = 1, 2, \ldots, d$ and summing both sides of the resulting inequalities over $i = 1, 2, \ldots, d$, we have

$$\left\| \nabla g(\mathbf{x}) - \nabla g_\phi(\mathbf{x}) \right\|_1 \quad \triangleq \quad \sum_{i=1}^{d} \left[ \left( \nabla g(\mathbf{x}) - \nabla g_\phi(\mathbf{x}) \right)^\top \mathbf{e}_i \right] \quad \leq \quad d \cdot \epsilon \; = \; \mathbf{O}(\epsilon) \tag{43}$$

Finally, we note that

$$\left\| \nabla g(\mathbf{x}) - \nabla g_\phi(\mathbf{x}) \right\|_2 \quad \leq \quad \left\| \nabla g(\mathbf{x}) - \nabla g_\phi(\mathbf{x}) \right\|_1 \quad \leq \quad \mathbf{O}(\epsilon) \tag{44}$$

which completes our proof and asserts that the gradient gap is indeed bounded by the Lipschitz constant of the function gap $h(\mathbf{x}) = g(\mathbf{x}) - g_\phi(\mathbf{x})$, which decreases as we optimize the loss function.

## B   TRAINING AND EVALUATION DETAILS OF MATCH-OPT

We use a feed-forward neural network with 4 layers ($512 \to 128 \to 32 \to 1$) activated by the Leaky ReLU function as the surrogate model for MATCH-OPT. For each task, we train the model using Adam optimizer (w/ fixed learning rate: 1e-4) for 200 epochs with a batch size of 128.

During the evaluation, we employ gradient updates for 150 iterations uniformly across all the tasks. This evaluation procedure uses an Adam optimizer with a learning rate fixed to 0.01 for all discrete tasks and 0.001 for all continuous tasks. We chose a larger learning rate for discrete tasks since the discrete inputs are converted into logits (same as all baselines).

## C   MATCH-OPT ABLATION WITHOUT REGRESSION REGULARIZER

In this section, we demonstrate the effectiveness of our practical consideration mentioned in Section 4. Specifically, we conduct an ablation study comparing two versions of MATCH-OPT using the original gradient matching loss in equation 11 (referred as MATCH-OPT (no-regularizer)) and an augmented version with regression regularizer along a set of sampled synthetic input sequences in equation 13 (referred to as MATCH-OPT (with-regularizer)). Table 3 and 4 below reports the performance of these ablated methods at the $100^{\text{th}}$ and $50^{\text{th}}$ percentile of solutions respectively. Overall, we observe that MATCH-OPT (with-regularizer) outperforms MATCH-OPT (no-regularizer) on 4/6 tasks for both the $100^{\text{th}}$-percentile and $50^{\text{th}}$-percentile metric, thus confirming that it is important to prioritize optimizing the gradient matching loss along critical trajectories of inputs.

## D   MEAN AND STANDARD DEVIATION RESULTS

As mentioned in the evaluation methodology, we ran each method for 4 different runs. This section reports the mean results from Tables 1 and 2 along with the corresponding standard deviations.

| METHOD | ANT | DKITTY | HOPPER | SCON | TF8 | TF10 |
|---|---|---|---|---|---|---|
| **MATCH-OPT** (no-regularizer) | 0.924 | 0.945 | 1.172 | **0.739** | 0.941 | **0.954** |
| **MATCH-OPT** (with-regularizer) | **0.931** | **0.957** | **1.572** | 0.732 | **0.977** | 0.924 |

Table 3: Performance comparison between versions of `MATCH-OPT` with and without regression regularizer at the 100th performance percentile (i.e., maximum solution) generated by each method.

| METHOD | ANT | DKITTY | HOPPER | SCON | TF8 | TF10 |
|---|---|---|---|---|---|---|
| **MATCH-OPT** (no-regularizer) | 0.572 | 0.876 | 0.372 | **0.471** | 0.551 | **0.768** |
| **MATCH-OPT** (with-regularizer) | **0.611** | **0.887** | **0.393** | 0.439 | **0.594** | 0.720 |

Table 4: Performance comparison between versions of `MATCH-OPT` with and without regression regularizer at the 50th performance percentile (i.e., maximum solution) generated by each method.

## E  RANK DISTRIBUTION PLOTS

To further illustrate the reliability of `MATCH-OPT`, this section visualizes the entire rank distribution of the tested algorithm across different percentile level (i.e., 25, 50, 75 and 100). Overall, we observe that `MATCH-OPT` (colored in red) consistently achieves lower mean and standard deviation of performance across all datasets at every percentile level, as compared to that of other baselines. This observation corroborates previous results presented in the main text, and confirms our hypothesis regarding the robustness of `MATCH-OPT`.

## F  ADDITIONAL EXPERIMENTS

In addition to the results reported in the main text, we have also compared `MATCH-OPT` with three additional baselines, which include DDOM Krishnamoorthy et al. (2023a), BO-qEI Wilson et al. (2017) and BDI Chen et al. (2022). The results are reported in Table 7 and Table 8 below.

In both the 50-th and 100-th percentile settings, it appears MATCH-OPT outperforms DDOM in all tasks. Furthermore, the results also show that MATCH-OPT performs the best in 6 out of 12 cases (across both the 100-th and 50-th percentile settings) while BO-qEI only performs best in 1 out of 12 cases. BDI performs best in 5 out of 12 cases, runs out of memory in 2 out of 12 cases. Overall, MATCH-OPT appears to perform more stable than BDI and is marginally better than BDI. It is also more memory-efficient than BDI as it does run successfully in all cases, while BDI runs out of memory in 2 cases. MATCH-OPT also outperforms BO-qEI significantly in 11 out of 12 cases.

## G  RUNNING TIME

We also report the running time achieved by all tested algorithms below

All reported running times are in seconds. Our algorithm incurs more time than other baselines but its total running time is still affordable in the offline setting: 4785s = 1.32hr. We do, however, want to remark that such complexity comparison is only tangential to our main contribution. Our main focus is on building optimizer with better and more stable performance overall, even at an affordable increase of running time. Furthermore, we want to point out that as some of the baselines (such as BONET) use an entirely different model which has a different number of parameters than ours, the reported running times here might not be comparable on the same compute platform.

| METHOD | ANT | DKITTY | HOPPER |
|---|---|---|---|
| GA | $0.271 \pm 0.013$ | $0.895 \pm 0.013$ | $0.780 \pm 0.462$ |
| ENS-MEAN | $0.517 \pm 0.039$ | $0.899 \pm 0.010$ | $1.524 \pm 0.710$ |
| ENS-MIN | $0.536 \pm 0.031$ | $0.908 \pm 0.019$ | $1.42 \pm 0.645$ |
| CMA-ES | $\mathbf{0.974} \pm 0.556$ | $0.722 \pm 0.001$ | $0.620 \pm 0.151$ |
| MINS | $0.910 \pm 0.034$ | $0.939 \pm 0.003$ | $0.150 \pm 0.186$ |
| CBAS | $0.842 \pm 0.015$ | $0.879 \pm 0.002$ | $0.150 \pm 0.014$ |
| ROMA | $0.832 \pm 0.055$ | $0.880 \pm 0.008$ | $2.026 \pm 0.225$ |
| BONET | $0.927 \pm 0.002$ | $0.954 \pm 0.0001$ | $0.395 \pm 0.0002$ |
| COMS | $0.885 \pm 0.024$ | $0.953 \pm 0.016$ | $\mathbf{2.270} \pm 0.237$ |
| **MATCH-OPT** | $0.931 \pm 0.011$ (2) | $\mathbf{0.957 \pm 0.014}$ **(1)** | $1.572 \pm 0.322$ (3) |

| METHOD | SCON | TF8 | TF10 |
|---|---|---|---|
| GA | $0.699 \pm 0.054$ | $0.954 \pm 0.020$ | $0.966 \pm 0.026$ |
| ENS-MEAN | $0.716 \pm 0.065$ | $0.926 \pm 0.005$ | $\mathbf{0.968} \pm 0.019$ |
| ENS-MIN | $0.734 \pm 0.058$ | $0.959 \pm 0.052$ | $0.959 \pm 0.021$ |
| CMA-ES | $\mathbf{0.757} \pm 0.013$ | $\mathbf{0.978} \pm 0.007$ | $0.966 \pm 0.007$ |
| MINS | $0.690 \pm 0.024$ | $0.900 \pm 0.059$ | $0.759 \pm 0.031$ |
| CBAS | $0.659 \pm 0.086$ | $0.916 \pm 0.035$ | $0.928 \pm 0.013$ |
| ROMA | $0.704 \pm 0.032$ | $0.664 \pm 0.015$ | $0.820 \pm 0.014$ |
| BONET | $0.500 \pm 0.002$ | $0.911 \pm 0.005$ | $0.756 \pm 0.006$ |
| COMS | $0.565 \pm 0.012$ | $0.968 \pm 0.018$ | $0.873 \pm 0.053$ |
| **MATCH-OPT** | $0.732 \pm 0.003$ (3) | $0.977 \pm 0.004$ (2) | $0.924 \pm 0.038$ (6) |

Table 5: Comparing MATCH-OPT and other baselines based on the $100^{\text{th}}$ percentile of the solutions (i.e., maximum solution) generated by each method. Each cell shows the mean and standard deviation of the function values found by each method over 4 runs. The individual rank of our method is included next to its reported performance for each benchmark.

# H LIMITATION

One potential limitation of our approach in comparison to other baselines is that our gradient match algorithm learns from pairs of data points. Thus, the total number of training pairs it needs to consume grows quadratically in the number of offline data points. For example, an offline dataset with $N$ examples will result in a set of $O(N^2)$ training pairs for our algorithm, which increases the training time quadratically. However, an intuition here is that training pairs are not equally informative and, in our experiments, it suffices to get competitive performance by just focusing on pairs of data along the sampled trajectories with monotonically increasing objective function values. This allows us to keep training cost linearly with respect to $N$.

On the other hand, while it is true that none of the existing baselines (including our algorithm) outperform others on all tasks, we believe that at least on these benchmark datasets, our algorithm tends to perform most stably across all tasks, as measured by the mean averaged rank reported in each of our performance tables. This is a single metric that is computed based on the performance of all baselines across all tasks. The end-user can make a judgment based on such metrics. In practice, by looking at how existing baselines perform overall on a set of benchmark tasks that are similar to a target task, one can decide empirically which baseline is most likely to be best for the target task.

| METHOD | ANT | DKITTY | HOPPER |
|---|---|---|---|
| GA | $0.130 \pm 0.029$ | $0.742 \pm 0.012$ | $0.089 \pm 0.07$ |
| ENS-MEAN | $0.192 \pm 0.010$ | $0.791 \pm 0.019$ | $0.209 \pm 0.035$ |
| ENS-MIN | $0.190 \pm 0.006$ | $0.803 \pm 0.005$ | $0.166 \pm 0.052$ |
| CMA-ES | $-0.049 \pm 0.003$ | $0.482 \pm 0.171$ | $-0.033 \pm 0.006$ |
| MINS | $0.614 \pm 0.034$ | $0.889 \pm 0.004$ | $0.088 \pm 0.170$ |
| CBAS | $0.376 \pm 0.023$ | $0.757 \pm 0.005$ | $0.013 \pm 0.002$ |
| ROMA | $0.448 \pm 0.013$ | $0.760 \pm 0.028$ | $0.370 \pm 0.008$ |
| BONET | $\mathbf{0.620} \pm 0.003$ | $\mathbf{0.897} \pm 0.0001$ | $0.390 \pm 0.0002$ |
| COMS | $0.557 \pm 0.015$ | $0.879 \pm 0.001$ | $0.379 \pm 0.005$ |
| **MATCH-OPT** | $0.611 \pm 0.007$ (3) | $0.887 \pm 0.003$ (3) | $\mathbf{0.393} \pm 0.005$ (1) |
| METHOD | SCON | TF8 | TF10 |
| GA | $0.641 \pm 0.036$ | $0.510 \pm 0.055$ | $0.794 \pm 0.013$ |
| ENS-MEAN | $0.644 \pm 0.070$ | $0.529 \pm 0.030$ | $\mathbf{0.796} \pm 0.006$ |
| ENS-MIN | $\mathbf{0.672} \pm 0.042$ | $0.490 \pm 0.052$ | $0.794 \pm 0.008$ |
| CMA-ES | $0.590 \pm 0.012$ | $0.592 \pm 0.015$ | $0.786 \pm 0.009$ |
| MINS | $0.414 \pm 0.011$ | $0.420 \pm 0.009$ | $0.465 \pm 0.016$ |
| CBAS | $0.099 \pm 0.008$ | $0.442 \pm 0.038$ | $0.613 \pm 0.012$ |
| ROMA | $0.420 \pm 0.030$ | $0.560 \pm 0.104$ | $0.780 \pm 0.400$ |
| BONET | $0.470 \pm 0.004$ | $0.505 \pm 0.004$ | $0.465 \pm 0.002$ |
| COMS | $0.414 \pm 0.023$ | $\mathbf{0.652} \pm 0.108$ | $0.606 \pm 0.027$ |
| **MATCH-OPT** | $0.439 \pm 0.016$ (6) | $0.594 \pm 0.015$ (2) | $0.720 \pm 0.015$ (6) |

Table 6: Comparing `MATCH-OPT` and baselines based on $50^{\text{th}}$ percentile of the solutions (i.e., median solution) generated by each method. Each cell shows the mean and standard deviation of the function values found by each method over 4 runs. The individual rank of our method is included next to its reported performance for each benchmark.

| METHOD | ANT | DKITTY | HOPPER | SCON | TF8 | TF10 |
|---|---|---|---|---|---|---|
| **MATCH-OPT** | 0.931 | 0.957 | 1.572 | 0.732 | 0.977 | 0.924 |
| **DDOM** | 0.768 | 0.911 | -0.261 | 0.570 | 0.674 | 0.538 |
| **BDI** | 0.967 | 0.940 | 1.706 | 0.735 | 0.973 | OOM |
| **BO-qEI** | 0.812 | 0.896 | 0.528 | 0.576 | 0.607 | 0.864 |

Table 7: Performance comparison between versions of `MATCH-OPT` with DDOM, BO-qEI and BDI at the $100^{\text{th}}$ performance percentile (i.e., maximum solution).

| METHOD | ANT | DKITTY | HOPPER | SCON | TF8 | TF10 |
|---|---|---|---|---|---|---|
| **MATCH-OPT** | 0.611 | 0.887 | 0.393 | 0.439 | 0.594 | 0.720 |
| **DDOM** | 0.554 | 0.868 | -0.570 | 0.390 | 0.418 | 0.461 |
| **BDI** | 0.583 | 0.870 | 0.400 | 0.480 | 0.595 | OOM |
| **BO-qEI** | 0.568 | 0.883 | 0.360 | 0.490 | 0.439 | 0.557 |

Table 8: Performance comparison between versions of `MATCH-OPT` with DDOM, BO-qEI and BDI at the $50^{\text{th}}$ performance percentile (i.e., maximum solution).

| | OURS | BO-QEI | CMA-ES | ROMA | MINS | CBAS | BONET |
|---|---|---|---|---|---|---|---|
| TIME | 4785 | 111 | 3804 | 489 | 359 | 189 | 614 |

| | | GA | ENS-MEAN | ENS-MIN | DDOM |
|---|---|---|---|---|---|
| | TIME | 45 | 179 | 179 | 2658 |

Table 9: Total running time (seconds) of all tested baselines.

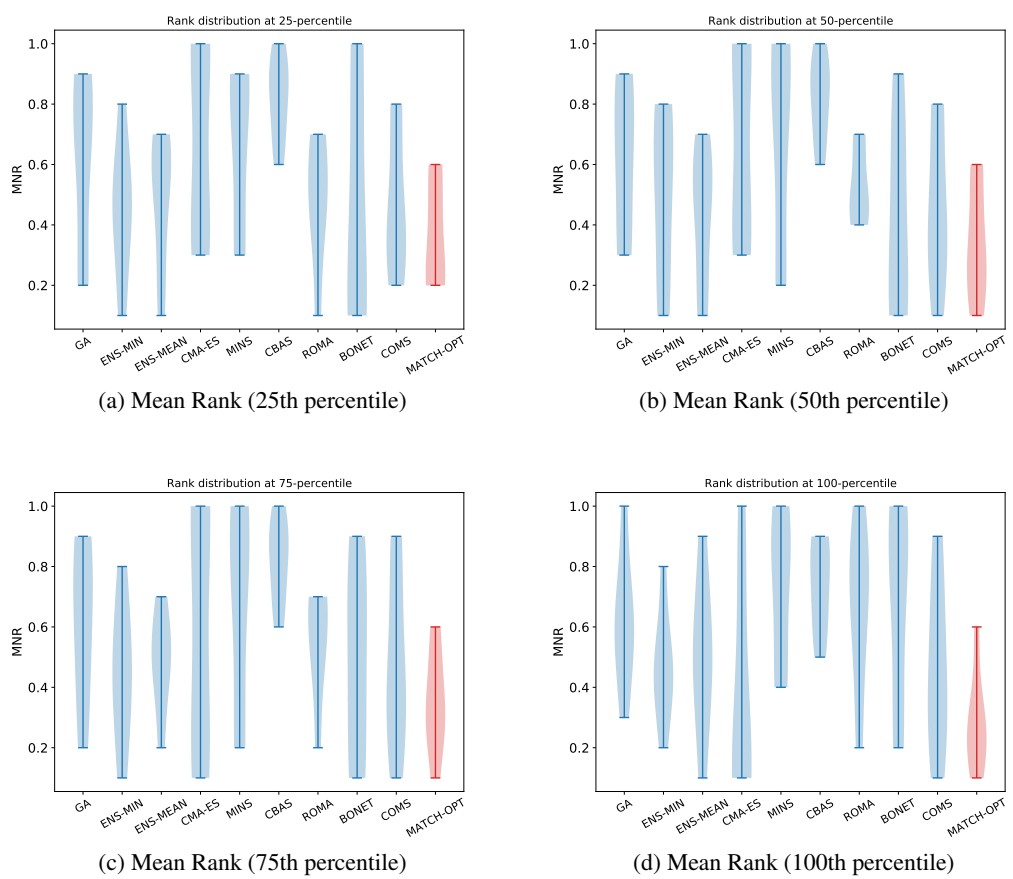

Figure 4: Plots of distributions of mean normalized rank (MNR) of the tested algorithms across all tasks at the (a) 25-th, (b) 50-th, (c) 75-th, and (d) 100-th performance percentile levels.

