# OpenReview forum: "Black-Box Gradient Matching for Reliable Offline Black-Box Optimization"
_ICLR.cc/2024/Conference — Submitted to ICLR 2024_

### Official Review · Reviewer_eLsu · 2023-10-21

**Soundness:** 1 poor
**Presentation:** 2 fair
**Contribution:** 1 poor
**Rating:** 5
**Confidence:** 5

**Summary:**

The paper addresses the challenge of offline design optimization in various scientific and engineering contexts, where physical or computational evaluations are costly, making real-time optimization impractical. The traditional solution has been to utilize surrogate models based on offline data to predict the objective function for unknown inputs. However, these surrogate models often show discrepancies from the true objective function, particularly outside the range of the offline data. This study introduces a novel theoretical framework to understand this discrepancy by looking at how well the surrogate models match the latent gradient of the true function. Based on this theoretical insight, the authors propose a new algorithm, MATCH-OPT, that aims to match the gradients of the oracle more closely. The efficacy of this approach is supported by experiments on real-world benchmarks, showing that MATCH-OPT outperforms existing methods in offline optimization tasks.

**Strengths:**

1. The concept of employing "fundamental line integration" as a technique for gradient estimation presents an intriguing approach.

2. The overall ranking performance of the method is commendable, underscoring its efficacy.

**Weaknesses:**

1. The paper frequently refers to "the gradient fields of the Oracle (i.e., the true objective function)." It's essential to note that a true objective function doesn't inherently possess a gradient. Continual mention of the oracle's gradient is foundational to this paper, and this assumption should be explicitly clarified by the author. I think this is the key drawback. If the author can clarify this point, I will increase my score.

2. The paper seems to overlook crucial baselines. It would be beneficial to reference and juxtapose the presented work against established benchmarks such as NEMO (https://arxiv.org/abs/2102.07970), CMA-ES, BO-qEI, BDI (https://arxiv.org/abs/2209.07507), and IOM (https://openreview.net/forum?id=gKe_A-DxzkH). These baselines are pivotal in this domain and warrant inclusion for a comprehensive analysis.

3. For better structuring, consider relocating the in-depth experimental results pertaining to the second question from the introduction to the exp section. This would make the introduction more concise and allow readers to delve into the specifics at the appropriate juncture.

**Questions:**

See Weaknesses.

---

> ### Author Response · Authors · 2023-11-15
> **Thank you for the review. We address all your concerns below.**
>
> **1. Assumption that the oracle has gradient.**
>
> We agree with the reviewer that not all oracle functions possess a gradient.
>
> **However, we want to point out that the implicit premise of the entire offline optimization field is that the developed techniques are meant for oracle functions which are continuous and differentiable. This premise is set the moment we set out to optimize the oracle using a differentiable surrogate such as the neural network.**
>
> This is the common theme in most (if not all) prior work.
>
> **This is also not an unreasonable assumption because attempting to optimize an oracle function with discontinuities is otherwise an ill-posed problem. Indeed, without the assumed differentiability, the oracle can be discontinued at any unseen data points at which its output behavior can be arbitrary, giving us no solid ground to relate between the training and (unseen) test output.**
>
> With this, we hope the reviewer would agree with us that we had in fact not used any assumptions prior work had not used, and consider re-evaluating the rating of our work.
>
> **2. Discussing and providing empirical comparisons with suggested baselines.**
>
> We would like to point out that we did compare with CMA-ES. We will include in our revision the following positioning with BO-qEI, BDI, NEMO, and IOM.
>
> **(1)** BDI uses forward and backward mappings to distill knowledge from the offline dataset to the design. BDI employs infinite width (neural tangent kernels) neural nets as surrogate models which might be limited in representation power for some applications.
>
> **(2)** IOM uses domain adaptation methods to enforce representation invariance between the training dataset and the distribution of optimized designs. Figuring out the right notion of distribution over optimized designs is an open challenge and the one used by IOM (points found during gradient ascent with a given surrogate) might not be the most effective one.
>
> **(3)** BO-qEI is standard Bayesian optimization (BO) run with a surrogate model as the objective function.
>
> **(4)** NEMO is an uncertainty quantification approach that uses normalized maximum-likelihood
> (NML) estimator which requires quantizing the output scores y. This quantization might be sub-optimal.
>
> Furthermore, we have also compared our method with BO-qEI & BDI and reported the results below.
>
> **100-th percentile**
>
> |           | Ant   | Dkitty | Hopper | Superconductor | tfbind-8 | tfbind-10 |
> |-----------|-------|--------|--------|----------------|----------|-----------|
> | BO-qEI    | 0.812 | 0.896  | 0.528  | 0.576          | 0.607    | 0.864     |
> | BDI       | **0.967** | 0.940  | **1.706**  | **0.735**          | 0.973    | OOM       |
> | MATCH-OPT | 0.931 | **0.957**  | 1.572  | 0.732          | **0.977**    | **0.924**     |
>
> **50-th percentile**
>
> |           | Ant   | Dkitty | Hopper | Superconductor | tfbind-8 | tfbind-10 |
> |-----------|-------|--------|--------|----------------|----------|-----------|
> | BO-qEI    | 0.568 | 0.883  | 0.360  | **0.490**          | 0.439    | 0.557     |
> | BDI       | 0.583 | 0.870  | **0.400**  | 0.480          | **0.595**    | OOM       |
> | MATCH-OPT | **0.611** | **0.887**  | 0.393  | 0.439          | 0.594    | **0.720**     |
>
> Note: OOM means out of memory.
>
> The results show that MATCH-OPT performs the best in 6 out of 12 cases (across both the 100-th and 50-th percentile settings) while BO-qEI only performs best in 1 out of 12 cases. BDI performs best in 5 out of 12 cases, runs out of memory in 2 out of 12 cases. Overall, MATCH-OPT appears to perform more stable than BDI and is marginally better than BDI. It is also more memory-efficient than BDI as it does run successfully in all cases, while BDI runs out of memory in 2 cases. MATCH-OPT also outperforms BO-qEI significantly in 11 out of 12 cases. These new results will be incorporated into our main text.
>
> In addition, we would also be happy to run comparative analysis with NEMO and IOM but the code for those baselines has not been released. We hope the reviewer would sympathize with us on this point since offline optimization is a fast-growing field and providing an exhaustive comparison with all prior work, including those with no release code is too difficult. Nonetheless, we have tried our best to compare with most prior work that provides released code.
>
> We hope the reviewer would take into account such practical difficulties and re-consider the rating of our work.
> We are looking forward to hearing back from you & will be happy to continue the discussion if you still have other questions for us.
>
> **3. Restructuring the experiment to make the introduction more concise.** We agree with the reviewer and will adjust the content accordingly in our revision.
>
> Thank you again for the detailed review & suggestions.

---

> > ### Comment · Reviewer_eLsu · 2023-11-16
> > **“Assumption that the oracle has gradient”**
> >
> > Could the authors list the works that explicitly assume task oracles are continuous & differentiable? According to the reviewer's understanding in this paper "Design-Bench: Benchmarks for Data-Driven Offline Model-Based Optimization", the oracles are generally black-box function and some tasks are even discrete.

---

> > > ### Author Response · Authors · 2023-11-17
> > > **Re: A (non-exhaustive) list of prior work that assumes oracle function has gradient**
> > >
> > > Thank you for the prompt response. We appreciate that you have given us the opportunity to elaborate more on this.
> > >
> > > As you requested, we list below **some of the state-of-the-art methods for offline optimization which explicitly assume task oracles are continuous & differentiable:**
> > >
> > > **1. COMS** -- https://arxiv.org/pdf/2107.0688 -- **this paper is published by the same group of authors on the pioneering paper “Design-Bench: Benchmarks for Data-Driven Offline Model-Based Optimization” referenced by the reviewer.**
> > >
> > > In page 5, 2nd column, the 2nd sentence of the paragraph before Proposition 1 states that
> > > “We assume that the true function f(x) is L-Lipschitz over the input space x”.
> > > By the Rademacher’s Theorem, such function is differentiable almost everywhere. This is therefore an explicit assumption that the function f(x) has gradient.
> > >
> > > **2. ROMA** -- https://arxiv.org/pdf/2110.14188
> > >
> > > In Section 2.4, the 4th sentence of the 1st paragraph states that “We train a proxy model f(z; θ) which operates on the latent space to approximate the true objective function, i.e. f ◦ g(x) ≈ f*(x)” where previously the true objective function is denoted by f*(x). Furthermore, both f and g are differentiable functions (which means their composition is also differentiable). We believe this is equivalent to an explicit assumption that the true objective function is differentiable.
> > >
> > > Furthermore, the central assumption of ROMA is the assumption of local smoothness as depicted in Figure 2. Smooth functions are differentiable.
> > >
> > > **3. BONET** -- https://arxiv.org/pdf/2206.10786
> > >
> > > In Proposition A2, it is stated that f is L-Lipschitz where f has been previously referred to as the oracle function in the 1st sentence of Section 2.1. Thus, similar to the explicit assumption COMS, this means the function is differentiable almost everywhere via the Rademacher theorem.
> > >
> > >
> > > More broadly, we want to emphasize again that **as long as a method uses a differentiable surrogate to model the objective function, it already implicitly assumes that the true objective function has gradient. This applies to at least the following prior works on offline optimization: BONET, DDOM, COMS, and ROMA.** Furthermore, **even for discrete spaces, all existing methods use logit based continuous relaxation. No method treats them directly in the original discrete space. This includes COMS, ROMA, MINS and others.**
> > >
> > > **Similar to these prior methods for offline optimization, we start with a differentiable surrogate model and an offline dataset to optimize an oracle function. Therefore, our work does not make any additional assumptions over the prior work**. I hope we have made this point clear in this response.
> > >
> > > Please let us know your thoughts. We are looking forward to hearing back from you.

---

> > > > ### Author Response · Authors · 2023-11-21
> > > > **Follow-up**
> > > >
> > > > Dear Reviewer eLSu,
> > > >
> > > > Thank you for taking the time to review and read our rebuttal. We appreciate that you had given us opportunity to provide further clarification regarding the gradient assumption.
> > > >
> > > > May we know if our response has addressed your remaining concern about the assumption?
> > > >
> > > > Thank you again and we are looking forward to hearing back from you soon.
> > > >
> > > > Best regards,
> > > >
> > > > Authors

---

> > > > > ### Comment · Reviewer_eLsu · 2023-11-21
> > > > > **Thanks for the author feedback**
> > > > >
> > > > > Thanks for the authors' feedback.
> > > > >
> > > > > I think there is a key difference between previous work and the authors regarding the assumption.
> > > > >
> > > > > COMs does not use the "oracle differentiable assumption" in its method design and only uses it for analysis. This also holds for BONET.
> > > > >
> > > > > As for ROMA, it only uses a differentiable NN to approximate the oracle, not saying the oracle is differentiable.
> > > > >
> > > > > "as long as a method uses a differentiable surrogate to model the objective function, it already implicitly assumes that the true objective function has gradient. " I do not think this is true. Even if the objective function does not have gradient (for example, has some discontinuities), we can still use NN to approximate it.
> > > > >
> > > > > In fact, there are many derivation-free baselines such as evolutionary algos and they are proposed based on the "derivation-free" assumption. Further, some tasks are not differentiable in this context. Take the TFBind8 as an example, and it uses logits as x and the y is the binding affinity.  Adding a small perturbation to x might change its original DNA sequence and leads to a sudden change of y. Regarding this point, the oracle is not differentiable.
> > > > >
> > > > > The reviewer will further discuss with other reviewers regarding this point.

---

> ### Author Response · Authors · 2023-11-21
> **Thank you for the quick follow-up**
>
> Thank you for the quick follow-up. From your response, you said that
>
> **COMs does not use the "oracle differentiable assumption" in its method design and only uses it for analysis. This also holds for BONET**
>
> This is true for us as well. If you take a look at Eq. (11), it does not involve the oracle's gradient anywhere. It only involves the gradient of the surrogate and the output difference between two sampled data points.
>
> Furthermore, you have also agreed that:
>
> **Even if the objective function does not have gradient (for example, has some discontinuities), we can still use NN to approximate it**
>
> This means it is fine to use the NN's gradient in place of the objective function's gradient (even if it does not exist). This is no different from our approach. So, once again, we do not use any assumptions that prior methods did not use.
>
> **We hope the reviewer could reconsider the rating given the above point**

---

> > ### Comment · Reviewer_eLsu · 2023-11-21
> > **Thanks for the feedback**
> >
> > Although Eq (11) does not use the oracle gradient, Eq (10) uses it for approximation.
> >
> > If I understand it correctly, COMs and BONET only use the assumption for further analysis, not using the assumption to derive the main method.  I did agree that "Even if the objective function does not have gradient (for example, has some discontinuities), we can still use NN to approximate it". In this case, you can not use the term "oracle gradient". You should say we have an ideal differentiable proxy which is the closest one to the ground-truth. This ideal proxy is what you call "oracle".
> >
> > Some task oracle in this context like TFB8 are not differentiable even if you use logits.
> >
> > I increase my score from 3 to 5. I think the paper still needs a lot to refine for publish. The authors should not only consider incorporating the above additional experiments but also should clarify the "oracle gradient" in the following version.

---

> ### Author Response · Authors · 2023-11-22
> **Thank you for the prompt response**
>
> Thank you for increasing the score.
>
> We understand that the reviewer’s suggestion is although (implementation-wise) we can use the NN’s gradient in place of the objective function’s gradient, it is better to write in the paper that
>
> **We have an ideal differentiable proxy which is the closest one to the ground-truth and this ideal proxy is what is called the oracle.**
>
> We have put this sentence in a new revision of our main text (see Section 2). This will solve the problem. The additional experiments in our original response will also be included. With this, we hope the reviewer would still reconsider the assessment that this paper needs a lot of refinement to publish.
>
> --
>
> In addition, we respectfully do **not** agree with the reviewer that COMS does not use the gradient assumption in its method design. It is written at the beginning of its Section 3 that
>
> **COMs learn estimates of the true function that do not overestimate the value of the ground truth objective on out-of-distribution inputs in the vicinity of the training dataset. As a result, COMs prevent erroneous overestimation that would drive the optimizer (Equation 2) to produce out-of-distribution inputs with low values under the ground-truth objective function**
>
> The last part is true only if its main theorem holds, which is under the assumption that the true function is L-Lipschitz, which implies the gradient assumption. In fact, **the regularizer term in the objective function -- see Eq.(4) -- is equivalently transformed to the more expressive form in Eq. (7) which is exactly what is bounded in the main theorem**. Thus, similar to our paper, the intriguing theoretical analysis of COMS informs its algorithm design in this case.
>
> What COMS assumed is fundamentally not different from the approximation step we used in Eq. (10). More importantly, we can remove the first equality in Eq. (10) so that there is no explicit writing of $\nabla g(\mathbf{x}) \simeq \nabla g_\phi(\mathbf{x})$ in the algorithm design and the rest of the technical narrative is still correct. But, we think the clarification statement above is already enough to make it clear.

---

### Official Review · Reviewer_1MFJ · 2023-10-30

**Soundness:** 2 fair
**Presentation:** 3 good
**Contribution:** 2 fair
**Rating:** 3
**Confidence:** 4

**Summary:**

In this manuscript, the authors propose an offline black-box optimization method by gradient matching.  In addition, the authors provide a bound of the difference between the function value at the solution of $m$-step true gradient update and that of $m$-step surrogate gradient update.

**Strengths:**

1. The proposed offline black-box optimization method via gradient matching is novel.

2.  The paper is well-written and well-organized.

**Weaknesses:**

1.  $\textbf{The optimum of the objective in Eq.(11) can not guarantee the learned gradient is close to the true gradient }$.

  Minimizing the term $(\Delta z - \Delta x ^\top \int _0^1\nabla g _\phi (tx + (1-t)x') dt)^2$  in Eq.(10) and Eq.(11) can not guarantee the gradient is close.   To be clear,  we have

\begin{equation}
(\Delta z - \Delta x ^\top \int _0^1\nabla g _\phi (tx + (1-t)x') dt)^2 = (\Delta x ^\top \int _0^1\nabla g  (tx + (1-t)x') dt - \Delta x ^\top \int _0^1\nabla g _\phi (tx + (1-t)x') dt)^2
\end{equation}
\begin{equation}
= ( \Delta x ^\top  \int _0^1 (\nabla g  (tx + (1-t)x') -  \nabla g _\phi (tx + (1-t)x') )dt   )^2
\end{equation}
The optimum is  $\Delta x ^\top  \int _0^1 (\nabla g  (tx + (1-t)x') -  \nabla g _\phi (tx + (1-t)x') )dt = 0$   not necessary $ \nabla g (\cdot)  = \nabla g _\phi (\cdot)  $.   Thus, when $\Delta x$ is orthogonal to $\int _0^1 (\nabla g  (tx + (1-t)x') -  \nabla g _\phi (tx + (1-t)x') )dt$, we get a trival  solution. The difference between $  \nabla g (\cdot)   $ and  $  \nabla g _\phi (\cdot)$ at the trivial solution can be arbitrarily large.

The true objective used in the manuscript is Eq.(13) instead of the gradient matching objective (11).  The objective (13) contains a standard regression objective term.   According to the above issue, the reviewer guesses that the regression objective term in Eq.(13) is still a key effective component.


2. $\textbf{The bound in Theorem 1 is very loose}$

The bound in Theorem 1 is trivial and loose, which exponentially grows w.r.t. the number of update steps $m$.  In addition, the term  $\max_{x}|| \nabla g(x) - \nabla g_\phi (x)  ||$ in Theorem 1 can grow to infinity.

3. $\textbf{The empirical improvement is not significant}$.

 The empirical results are not convincing enough to demonstrate the claimed advantage of the proposed method.  In Table 1 and Table 2, the proposed method does not consistently outperform other baseline methods.

4. $\textbf{Recent related baselines are missing}$.

A comparison with the recent related offline black-box optimization method [1] is missing.   The experimental setup in [1] is quite similar to this manuscript.

[1] Krishnamoorthy et al. Diffusion Models for Black-Box Optimization. ICML 2023

**Questions:**

Q1.  Could the authors address the concerns in the above section?

Q2.  Could the authors include more comparisons with the related baseline [1] ?

Q3. In this paper, the authors assume the search method given the learned surrogate is gradient ascent.  Does the gradient ascent search be the unique search method? Could the authors include additional comparisons with other search methods given the learned surrogate?

---

> ### Author Response · Authors · 2023-11-15
> **Thank you for the review. We address all your concerns below.**
>
> **1. Optimizing Eq. (11) might not guarantee gradient match.** Thank you for the question. We will prove below that **optimizing Eq. (11) does necessarily guarantee gradient match** to address this concern.
>
> First, we understand the reviewer's key concern here is that there might exist a parameter $\phi$ such that the integrated vector
>
> $\mathbf{F}_{\phi}(\mathbf{x},\mathbf{x}') \ = \ \int_0^1 \nabla g(t\mathbf{x} + (1-t)\mathbf{x}')\mathrm{d}t$
>
> $- \int_0^1 \nabla g_\phi(t\mathbf{x} + (1-t)\mathbf{x}')\mathrm{d}t$
>
> is non-zero but instead orthogonal to $\Delta\mathbf{x} = \mathbf{x} - \mathbf{x}'$. We can show that **such a parameter does not exist** in what follows.
>
> First, following your argument, for such a parameter phi, we have
>
> $\mathbf{F}_{\phi}(\mathbf{x}, \mathbf{x}’)^\top(\mathbf{x} - \mathbf{x}’) = 0$ for all $(\mathbf{x}, \mathbf{x}’)$
>
> Next, by the line integration theorem, we have:
>
> **(1)** $g(\mathbf{x}) - g(\mathbf{x}’) = (\mathbf{x} - \mathbf{x}’)^\top \int_0^1 \nabla g(t\mathbf{x} + (1-t)\mathbf{x}’)\mathrm{d}t$
>
> **(2)** $g_\phi(\mathbf{x}) - g_\phi(\mathbf{x}’) = (\mathbf{x} - \mathbf{x}’)^\top \int_0^1 \nabla g_\phi(t\mathbf{x} + (1-t)\mathbf{x}’)\mathrm{d}t$
>
> for all $(\mathbf{x}, \mathbf{x}’)$. As such, combining **(1)** and **(2)** leads to
>
> $\mathbf{F}_\phi(\mathbf{x}, \mathbf{x}’)^\top(\mathbf{x} - \mathbf{x}’) = $
>
> $g(\mathbf{x}) - g(\mathbf{x}’) - g_\phi(\mathbf{x}) + g_\phi(\mathbf{x}’)$
>
> Or equivalently, $g(\mathbf{x}) - g(\mathbf{x}’) - g_\phi(\mathbf{x}) + g_\phi(\mathbf{x}’) =  0$ since $\mathbf{F}_\phi(\mathbf{x}, \mathbf{x}’)^\top(\mathbf{x} - \mathbf{x}’) = 0$ for all $(\mathbf{x}, \mathbf{x}’)$.
>
> We can thus rewrite the above as $g(\mathbf{x}) - g_\phi(\mathbf{x}) = g(\mathbf{x}’) - g_\phi(\mathbf{x}’)$ for all $(\mathbf{x}, \mathbf{x}’)$, which means there must exist a constant $c$ such that $g(\mathbf{x}) - g_\phi(\mathbf{x}) = c$ for all $\mathbf{x}$.
>
> Taking gradient with respect to $\mathbf{x}$ on both sides leads to
>
> $\nabla g(\mathbf{x}) - \nabla g_\phi(\mathbf{x}) = 0$ or equivalently, $\nabla g(\mathbf{x}) = \nabla g_\phi(\mathbf{x})$
>
> This means for any parameter $\phi$ such that
>
> $\mathbf{F}_\phi(\mathbf{x}, \mathbf{x}’)^\top(\mathbf{x} - \mathbf{x}’) = 0$
>
> for all $(\mathbf{x}, \mathbf{x}')$, the gradients of $g(\mathbf{x})$ and $g_\phi(\mathbf{x})$ are guaranteed to match.
>
> Thus, **the case that the reviewer is concerned about does not exist**. Please let us know if this has addressed your concern.
>
> --
>
> Furthermore, please also refer to Appendix C where we show an ablation study between the  training objective with and without the regression term. This addresses the concern of the reviewer that the performance might come solely from the regression term. This is not true according to our experiment in Appendix C: including the regression term improves the performance as expected but even without it, the performance is still very competitive, asserting the impact of the gradient match term, which matches with the implication of Theorem 1.
>
> **2. The bound in Theorem 1 seems to grow exponentially in $m$ and hence, might be loose.** Thank you for another critical question. We understand the concern here is that the bound seems to grow exponentially in the no. of iterations $m$, and might be loose as a result. To address this concern, **we want to assure the reviewer that the bound will NOT grow exponentially in the no. of iterations with a standard choice of the learning rate  $\lambda$**. Here is how.
>
> Let us choose $\lambda$ such that it is no more than $1/m$, which is the case in all our experiments where $m = 200$ and $\lambda = 10^-4$, as stated in the appendix. With this choice, we have:
>
> $(1 + \lambda \cdot \mu)^{(m - 1)} \leq (1 + \mu / m)^{(m-1)} < (1 + \mu/m)^m$
>
> which will approach $e^mu$ in the limit of $m$. Here, we use the known fact that $\mathrm{lim}_{m\rightarrow\infty}(1 + \mu/m)^m = e^\mu$ with $\mu > 0$.
>
> As such, when $m$ is sufficiently large the bound in Theorem 1 is upper-bounded with
>
> $m \cdot\lambda \cdot \ell \cdot (1 + \lambda \cdot \mu)^{(m - 1)} \cdot \text{gradient-gap} \simeq \ell \cdot e^\mu \cdot\text{gradient-gap}$
>
> which asserts that the worst-case performance gap of our offline optimizer is approaching (in the limit of $m$) $\ell \cdot e^\mu \cdot \text{gradien-gap} = \mathbf{O}(\text{gradient-gap})$ which is not dependent on the number of gradient steps.
>
> Thus, the bound is not growing exponential in $m$ and hence, it is not loose and trivial. Please let us know if this has addressed the reviewer’s concern.
>
>
> **We will address your remaining concerns in the next comment**

---

> > ### Author Response · Authors · 2023-11-15
> > **Response to remaining concerns (cont.)**
> >
> > **3. The empirical improvement is not significant.** We respectfully disagree with the statement that the empirical improvement is not significant because our method does not outperform all other methods in all individual cases.
> >
> > We believe it is not a fair statement given that none of the existing methods (including DDOM [1]) has superior empirical improvement over all other methods in all individual cases. For example, DDOM outperforms all other methods in 2/6 tasks but argues that its average mean rank is better. It is thus only fair for MATCH-OPT to claim empirical significance using the exact same argument: Its average mean rank is better than all others in both Tables 1 and 2.
> >
> > In addition, we note that both Reviewer **eLsu** and Reviewer **7Vbx** also agrees on the significance our empirical improvement:
> >
> > **eLsu**: "The overall ranking performance of the method is commendable, underscoring its efficacy"
> >
> > **7Vbx**: "The algorithm performs well over baselines both on Mean Normalized Rank and Mean normalized score metrics for both 50 percentile and 100 percentile."
> >
> > Furthermore, Reviewer **ehny** also gave us a high overall score, which (we believe) also indicates agreement with our substantial empirical improvement.
> >
> > We hope the reviewer would reconsider the original assessment of our empirical work given the above.
> >
> > **4. Comparison with DDOM.** As requested, we have run a thorough comparison with DDOM, which is reported below.
> >
> > **100-th percentile**
> >
> > |           | Ant   | Dkitty | Hopper | Superconductor | tfbind-8 | tfbind-10 |
> > |-----------|-------|--------|--------|----------------|----------|-----------|
> > | DDOM      | 0.768 | 0.911  | -0.261 | 0.570          | 0.674    | 0.538     |
> > | MATCH-OPT | **0.931** | **0.957**  | **1.572**  | **0.732**          | **0.977**    | **0.924**     |
> >
> > **50-th percentile**
> >
> > |           | Ant   | Dkitty | Hopper | Superconductor | tfbind-8 | tfbind-10 |
> > |-----------|-------|--------|--------|----------------|----------|-----------|
> > | DDOM      | 0.554 | 0.868  | -0.570 | 0.390          | 0.418    | 0.461     |
> > | MATCH-OPT | **0.611** | **0.887**  | **0.393**  | **0.439**          | **0.594**    | **0.720**     |
> >
> > In both the 50-th and 100-th percentile settings, it appears MATCH-OPT outperforms DDOM in all tasks. For better clarity, we will also incorporate these results into the revised manuscript.
> >
> > We used the source code from the DDOM GitHub repository provided here https://github.com/siddarthk97/ddom.
> >
> > **5. Does the gradient ascent search be the unique search method?**
> >
> > Currently, our theory has been developed for gradient ascent, which shows substantial improvement over a diverse set of baselines that employ a diverse range of search strategies. Incorporating new search methods into our framework is highly non-trivial, which requires a significant generalization of Theorem 1. This would deserve a separate treatment in another follow-up work.
> >
> > We thank the reviewer for the suggestion and we are happy to investigate this further but as highlighted above, the complexity of this investigation is way beyond the scope of a rebuttal. It is also tangentially related to our main contribution here.
> >
> > To this end, we also note that most existing approaches in offline optimization are often based on a single search strategy. For example, one of the pioneering works in this field, COMS, is based on gradient ascent alone.

---

> ### Comment · Reviewer_1MFJ · 2023-11-16
>
> Thanks for the authors' detailed response.   Part of my concerns have been addressed. I am happy to see the additional insight provided by the authors. However, some of my concerns remain unsolved.
>
> $1. \textbf{Optimizing Eq. (11) might not guarantee gradient match}$
>
> It seems that the authors assume  $\mathbf{F}_{\phi}(\mathbf{x}, \mathbf{x}’)^\top(\mathbf{x} - \mathbf{x}’) = 0$ for all $(\mathbf{x}, \mathbf{x}’)$.
>
> However,  we may only guarantee  $\mathbf{F}_{\phi}(\mathbf{x}, \mathbf{x}’)^\top(\mathbf{x} - \mathbf{x}’) = 0$ for finite $(\mathbf{x}, \mathbf{x}’)$ in the training set, even if we assume a strong enough optimizer.
>
> As a result, we only know $g(\mathbf{x}) - g_\phi(\mathbf{x}) = c$ for finite samples $\mathbf{x}$ in the training set instead of $\forall \mathbf{x} \in \mathbb{R}^d$.
>
> Thus, we may not conclude $\nabla g(\mathbf{x}) - \nabla g_\phi(\mathbf{x}) = 0$ for all $\mathbf{x}$, especially for the high-dimensional cases that the training data is too sparse to cover the entire domain.

---

> ### Author Response · Authors · 2023-11-20
> **Thank you for the quick follow-up. The gradient gap is either zero or bounded as detailed below.**
>
> Thank you for the prompt response. Previously, we had proved that in the limit of data if we can find the true optimal solution that zeroes out the generalized loss, the gradient is matched.
>
> Now, suppose that $\mathbf{F}_\phi(\mathbf{x}, \mathbf{x}’)^\top(\mathbf{x} - \mathbf{x}’)$ is not zero everywhere (i.e., the generalized loss is not zero), we will show that the gradient gap is still bounded by a decreasing quantity.
>
> To see this, note that $\mathbf{F}_\phi(\mathbf{x}, \mathbf{x}’)^\top(\mathbf{x} - \mathbf{x}’)$
>
> $= (g(\mathbf{x}) - g_\phi(\mathbf{x})) - (g(\mathbf{x}’) - g_\phi(\mathbf{x}’))$
>
> which was established in the 5th equation of our previous message.
>
> Let $h(\mathbf{x}) \triangleq g(\mathbf{x}) - g_\phi(\mathbf{x})$. It follows that $|\mathbf{F}_\phi(\mathbf{x}, \mathbf{x}’)^\top(\mathbf{x} - \mathbf{x}’)| = |h(\mathbf{x}) - h(\mathbf{x}')|$ which means our loss function is working towards minimizing $|h(\mathbf{x}) - h(\mathbf{x}')|^2$ over $(\mathbf{x},\mathbf{x}')$. Intuitively, this will make $h(\mathbf{x})$ smoother as the output distance between different inputs are being reduced.
>
> As a result, this process will reduce the Lipschitz constant $\epsilon$ of $h(.)$. Here, we are alluding to the hypothesis that if a model behaves smoothly on the training data, it will also be smooth on the test data. We believe this is reasonable in the broader context as well as this context because most work in offline optimization present some forms of smoothing out the behavior of the model on test data and it is often achieved via conditioning the model on the training data.
>
> Now, **we will show that the gradient gap $||\nabla g(\mathbf{x}) - \nabla g_\phi(\mathbf{x})||$ is bounded by $\epsilon$:**
>
> First, by definition, $|h(\mathbf{x}) - h(\mathbf{x}')| \leq \epsilon \cdot ||\mathbf{x} - \mathbf{x}'||$ which implies
>
> $|(g(\mathbf{x}) - g(\mathbf{x}')) - (g_\phi(\mathbf{x}) - g_\phi(\mathbf{x}'))| \leq \epsilon ||\mathbf{x} - \mathbf{x}'||$  -- due to the definiton of $h(.)$
>
> Dividing both sides by $||\mathbf{x} - \mathbf{x}'||$ leads to:
>
> $\left|\frac{(g(\mathbf{x}) - g(\mathbf{x}'))}{||\mathbf{x} - \mathbf{x}'||} - \frac{(g_\phi(\mathbf{x}) - g_\phi(\mathbf{x}'))}{||\mathbf{x} - \mathbf{x}'||}\right| \leq \epsilon $
>
> Since this holds for any $(\mathbf{x}, \mathbf{x'})$, we can choose $\mathbf{x'} = \mathbf{x} + t\cdot \mathbf{e_j}$, where $t$ is some scalar and $\mathbf{e_j}$ denotes the $d$-dimensional one-hot vector with the hot component at the $j$-th coordinate. Here, $d$ denotes the input dimension. This gives:
>
> $-\epsilon \leq \frac{(g(\mathbf{x} + t \cdot \mathbf{e_j} ) - g(\mathbf{x}))}{t||\mathbf{e_j}||} - \frac{(g_\phi(\mathbf{x} + t \cdot \mathbf{e_j}) - g_\phi(\mathbf{x}))}{t||\mathbf{e_j}||} \leq \epsilon $
>
> Now, taking $\mathrm{lim}_{t\rightarrow 0}$ on all terms in the inequality and using the definition of directional gradient, we will arrive at:
>
> $-\epsilon \leq (\nabla g(\mathbf{x}) - \nabla g_\phi(\mathbf{x}))^\top \mathbf{e_j} \leq \epsilon$ , or $\left| (\nabla g(\mathbf{x}) - \nabla g_\phi(\mathbf{x}))^\top \mathbf{e_j} \right| \leq \epsilon $
>
> For each value of $j = 1, 2, \ldots, d$ , we will obtain one such inequality. Summing all these inequalities gives:
>
> $||\nabla g(\mathbf{x}) - \nabla g_\phi(\mathbf{x})||_1 \leq \epsilon \cdot d = \mathbf{O}(\epsilon) $
>
> Finally, since norm-2 is upper bounded by norm-1, we also have $||\nabla g(\mathbf{x}) - \nabla g_\phi(\mathbf{x})||_2 \leq \epsilon $.
>
> Thus, **the gradient gap will be bounded by the Lipschitz constant of the function gap $h(\mathbf{x}) = g(\mathbf{x}) - g_\phi(\mathbf{x})$ and this constant decreases as we optimize the loss function**. In addition, if $\mathbf{x}$ is within the training data, the gradient gap zeros out as established previously.
>
> **Please let us know if this has addressed your concern. Thank you again for keeping this discussion alive.**

---

> > ### Author Response · Authors · 2023-11-23
> > **Quick follow-up**
> >
> > Dear Reviewer,
> >
> > Thank you again for the discussion so far.
> >
> > Have you got a chance to look at the bound that we derived for the gradient gap when the optimization is imperfect?
> >
> > We are interested to know if this has addressed your remaining concern. You can find a more detailed version of it at the end of Appendix A.
> >
> > Best regards,
> >
> > Authors

---

> ### Comment · Reviewer_1MFJ · 2023-11-24
>
> Thanks for the authors' detailed response.  However, my concern has not been addressed.
>
>
> **(1). Finite Sample Case**
>
> **There exists $g_\phi$ that achieves zero loss on the finite training dataset $\mathfrak{D}$, but the gradient gap is large even on the training samples.**
>
> Consider a finite dataset $ \mathfrak{D}:= \\{ +1,-1 \\}^d \subset \mathbb{R}^d$,
>
> The loss in Eq.(11) on $  \mathfrak{D}   $ is given below:   (Note that $\mathfrak{D}$ denotes the training dataset under Eq.(11) in the paper )
>
> \begin{equation}
> \mathfrak{L}(\phi) =  \sum _{\boldsymbol{x}, \boldsymbol{x}' \in \\{ +1,-1 \\}^d } ( g(\boldsymbol{x})- g _\phi (\boldsymbol{x}) - (g(\boldsymbol{x}')- g _\phi (\boldsymbol{x}') ) )^2  = \sum _{\boldsymbol{x}, \boldsymbol{x}' \in \\{ +1,-1 \\}^d } ( h(\boldsymbol{x}) -  h(\boldsymbol{x}') )^2
> \end{equation}
>
> A solution that achieves zero loss $\mathfrak{L}(\phi) =0$ on the training set $\\{ +1,-1 \\}^d$ is $h(\boldsymbol{x})= g(\boldsymbol{x}) - g _\phi (\boldsymbol{x}) = c $ for $ \forall \boldsymbol{x} \in \\{ +1,-1 \\}^d$, where $c$ denotes a constant.
>
> However,  there exist functions $h(\boldsymbol{x})$ that achieves zero loss $\mathfrak{L}(\phi) =0$ on the training set $\\{ +1,-1 \\}^d$, but the gradient gap is large even on the training samples.
>
> For example, a simple function $h(\boldsymbol{x})= (\boldsymbol{x}^\top\boldsymbol{x} )^{5}$.  For $ \forall \boldsymbol{x} \in \\{ +1,-1 \\}^d$, we have $h(\boldsymbol{x})= (\boldsymbol{x}^\top\boldsymbol{x} )^{5} = d^{5}$. Thus, the loss is zero $\mathfrak{L}(\phi) =0$.
>
> But the gradient of $h(\boldsymbol{x})$ on any training sample $\boldsymbol{x}$ is $\nabla h(\boldsymbol{x}) = 10(\boldsymbol{x}^\top\boldsymbol{x} )^4  \boldsymbol{x}$.
>
> The gradient gap on the training sample $ \forall \boldsymbol{x} \in \\{ +1,-1 \\}^d$  is
>
> \begin{equation}
> \\| \nabla g(\boldsymbol{x}) - \nabla g _\phi (\boldsymbol{x})    \\|_2^2 = \\| \nabla h(\boldsymbol{x})    \\|_2^2 = 100 (\boldsymbol{x}^\top\boldsymbol{x} )^{8} \\|  \boldsymbol{x}    \\|_2^2= 100 d^{9}
> \end{equation}
>
> For a small $d=100$,  we have the gradient gap is
> \begin{equation}
> \bf{ \\| \nabla g(\boldsymbol{x}) - \nabla g _\phi (\boldsymbol{x})    \\|_2^2 = 100 ^{10} =  10 ^{20} }
> \end{equation}
>
>
>
>
> In fact,  minimizing loss Eq.(11) on the finite training dataset $\mathfrak{D}$ can not guarantee the local variation, especially for high-dimensional cases.
>
> For the above example with a small $d=100$,  even with $2^{100}$ training samples,  the gradient gap $\\| \nabla g(\boldsymbol{x}) - \nabla g _\phi (\boldsymbol{x})    \\|_2^2$ on the training sample can grows up to $\bf{10 ^{20}}$.
>
>
>
>
> **(2) Infinite sample cases**
>
> The key assumption of the  infinite sample cases is that we can achieve a constant function  $h(\boldsymbol{x})=c$ for $\forall \boldsymbol{x} \in \mathbb{R}^d$    to ensure that   the gradient  $\nabla h(\boldsymbol{x}) = \boldsymbol{0}$.  However, this assumption may be strong and impractical.

---

### Official Review · Reviewer_ehny · 2023-10-31

**Soundness:** 3 good
**Presentation:** 4 excellent
**Contribution:** 4 excellent
**Rating:** 8
**Confidence:** 4

**Summary:**

This work provide a solid investigation on important problem in offline black-box optimization. The paper is full of insights and is enjoyable to read.

**Strengths:**

The paper is full of insights and is enjoyable to read.

**Weaknesses:**

I am satisfied with current version.

**Questions:**

1. Could you share more insight on why organizing training data into monotonically increasing trajectories is able to mimic optimization paths? In particular, could you shed more light on the equation (13)?

---

> ### Author Response · Authors · 2023-11-15
> **Thank you for recognizing our contribution. We answer your question below**
>
> Thank you for the strong support of our work. We are very happy to know that you find our work insightful!
>
> To answer your question, organizing training data into monotonically increasing trajectories encourages the model to learn the behavior of a gradient-based optimization algorithm, which in fact produces (mostly) monotonically increasing trajectories. This allows the gradient matching algorithm to focus more on strategic input pairs that are more relevant for gradient estimation.
>
> Furthermore, Eq. (13) combines both the gradient match loss and the regression loss. The regression loss helps as a regularizer that amplifies the importance of accurate gradient matching along the sampled trajectories which encodes our bias that the optimal trajectories would be sufficiently close to one of those sampled trajectories.

---

### Official Review · Reviewer_7Vbx · 2023-11-01

**Soundness:** 3 good
**Presentation:** 3 good
**Contribution:** 3 good
**Rating:** 6
**Confidence:** 2

**Summary:**

The paper first provides a bound for the performance gap between oracle and a chosen surrogate as a function of how well the surrogate matches the gradient field of the oracle on the offline training data and then using the analysis done to bound the performance gap formulates an algorithm which uses multiple monotonic trajectories of hopping over training points after binning and splitting them based on percentile values for the task of black-box offline optimization where access to oracle is unavailable in the sense that no new input and output value cannot be acquired or sampled. The authors show that the algorithm does better compared to other baselines on tasks and datasets proposed in Trabucco et al., 2022 both in terms of Mean Normalized Rank and Mean Normalized score over multiple percentiles of candidate solutions provided by the algorithms.

**Strengths:**

1. The paper is mostly well written and the relevant literature and references are covered well.
2. The math looked sound to me as far as I could see.
3. The algorithm performs well over baselines both on Mean Normalized Rank and Mean normalized score metrics for both 50 percentile and 100 percentile.
4. The paper gives complexity analysis of the proposed algorithm.
5. Figures look good and support the narrative and many(9) baseline algorithms are tried and compared with the proposed algorithm.

**Weaknesses:**

I am not familiar with this field of research and so my comments should be taken with a grain of salt. I am ready to revise the score after reading other reviews and authors' rebuttal.

1. Typo: Page 9, missing reference.
2.  There is some repetition in the section Evaluation Methodology and section on Results and Discussion, the readibility of those sections can be improved.
3. The paper does not compare the memory and time complexity of the algorithm with baselines. This will also depend on the hyperparameters of the optimization algorithm like the discretization parameter.
4. Some questions are unanswered which I list below.
5. The paper does not explicitly state its limitations compared to baselines especially since no baseline and proposed algorithm consistently outperforms the other methods on all tasks. Can the end-user make a judgment ?

**Questions:**

Some questions, the answers to which can help the paper
1. How does the discretization parameter affect the performance of the algorithm ?
2. How do the optimization hyperparameters affect the performance, also how do you choose or tune $\alpha$, the term balancing the two loss terms ?
3. Right now, the objective contains a sum of two terms: value matching loss and gradient matching loss as shown in Fig. 2, what effect does each term have and are they of the same scale or their scales vary a lot on these tasks ?

---

> ### Author Response · Authors · 2023-11-17
> **Thank you for the review and favorable initial rating. We would like to address your question below**
>
> **1. Compare running time with other baselines.** Thank you again for the suggestion. We will quote the memory and time complexity of the baselines in the revised version. Some of these are not made explicitly in their corresponding paper but we can look into their formulation to deduce their complexities.
>
> We do, however, want to remark that such complexity comparison is only tangential to our main contribution. Our main focus is on building optimizer with better and more stable performance overall, even at an affordable increase of running time. Furthermore, we want to point out that as some of the baselines (such as BONET) use an entirely different model which has a different number of parameters than ours, the complexity comparison might not be apple-to-apple.
>
> But regardless, we want to report the averaged running time of the baselines below.
>
> |      | OUR  | BO-qEI | CMA-ES | ROMA | MINS | CBAS | BONET | GA | ENS-MEAN | ENS-MIN | DDOM |
> |------|------|--------|--------|------|------|------|-------|----|----------|---------|------|
> | Time | 4785 | 111    | 3804   | 489  | 359  | 189  | 614   | 45 | 179      | 179     | 2658 |
>
> All reported running times are in seconds. Our algorithm incurs more time than other baselines but its total running time is still affordable in the offline setting: 4785s = 1.32hr.
>
> Once again, **we want to emphasize that (1) it is not our claim contribution to be faster than the baseline; and (2) as we are optimizing expensive-to-evaluate objective functions (e.g., evaluating candidate materials through physical experiments), providing more stable performance (i.e., MNR) in exchange for an affordable increase (around 1hr) of offline computational time is still preferred by the practitioners.**
>
> **2. Limitations compared to baselines.** One potential limitation of our approach in comparison to other baselines is that our gradient match algorithm learns from pairs of data points. Thus, the total number of training pairs it needs to consume grows quadratically in the number of offline data points. For example, an offline dataset with N examples will result in a set of O(N^2) training pairs for our algorithm, which increases the training time quadratically. However, an intuition here is that training pairs are not equally informative and, in our experiments, it suffices to get competitive performance by just focusing on pairs of data along the sampled trajectories with monotonically increasing objective function values. This allows us to keep training cost linearly with respect to N.
>
> On another note, while it is true that none of the existing baselines (including our algorithm) outperform others on all tasks, we believe that at least on these benchmark datasets, our algorithm tends to perform most stably across all tasks, as measured by the mean averaged rank reported in each of our performance tables. This is a single metric that is computed based on the performance of all baselines across all tasks. The end-user can make a judgment based on such metrics. In practice, by looking at how existing baselines perform overall on a set of benchmark tasks that are similar to a target task, one can decide empirically which baseline is most likely to be best for the target task.
>
> **3.How does the discretization parameter affect the performance of the algorithm?** Our empirical inspections suggest that a discretization $\kappa = 5$ in Eq. 12 is sufficient to get good performance. Further increasing $\kappa$ only results in negligible changes in performance. In addition, increasing kappa will increase the time complexity linearly, as detailed in our Complexity Analysis paragraph following Eq. (13).
>
> **We will address your remaining questions in the next comment.**

---

> > ### Author Response · Authors · 2023-11-17
> > **Rebuttal (cont.)**
> >
> > **4. How do you choose or tune $alpha$, the term balancing the two loss terms?** We set $\alpha = 1$ in all our experiments since the regression and gradient match terms have the same unit scale. Both can be re-interpreted in terms of output differences. For the gradient match term, the line integration theorem implies:
> >
> > (1) $g(\mathbf{x}) - g(\mathbf{x}’) = (\mathbf{x} - \mathbf{x}’)^\top \int_0^1 \nabla g(t\mathbf{x} + (1-t)\mathbf{x}’)\mathrm{d}t$
> >
> > (2) $g_\phi(x) - g_\phi(\mathbf{x}’) = (\mathbf{x} - \mathbf{x}’)^\top \int_0^1 \nabla g_\phi(t\mathbf{x} + (1-t)\mathbf{x}’)\mathrm{d}t$
> >
> > for all $(\mathbf{x}, \mathbf{x}’)$. As such, combining (1) and (2) allows us to re-interpret the gradient loss in the summation form of $(g(\mathbf{x}) - g(\mathbf{x}’) - g_\phi(\mathbf{x}) + g_{\phi}(\mathbf{x}’))^2$ over multiple pairs of $(\mathbf{x}, \mathbf{x}’)$. This is clearly in the same unit scale as the regression loss (regardless of the task).
> >
> > **5. What effect do value matching loss and gradient matching loss have and are they of the same scale?** The two terms in our objective function are indeed on the same scale, regardless of the matching loss. This is due to the fact that by the line integration theorem, the gradient match loss can be re-interpreted as the difference between corresponding function outputs, which is incorporated directly into the regression loss. We have shown this in the answer to your previous question.
> >
> > The effect of the terms are as follows: (1) the gradient match loss  helps in minimizing the upper-bound of the offline optimization quality as  shown by Theorem 1; and (2) the regression loss on the other hand acts as a regularizer that amplifies the importance of accurate gradient matching along the sampled trajectories which encodes our bias that the optimal trajectories would be sufficiently close to one of those sampled trajectories.
> >
> > **6. Other suggestions on including missing reference & improving readability of Evaluation Methodology, Results and Discussion sections** Thank you for the suggestions. We will revise our paper according to your feedback.
> >
> > **We hope the above has addressed all your questions. Otherwise, please let us know if you still have additional questions for us. We will be very happy to continue the discussion. Thank you very much for your detailed review!**

---

> > > ### Author Response · Authors · 2023-11-21
> > > **Follow-up**
> > >
> > > Dear Reviewer 7Vbx,
> > >
> > > Thank you very much for the detailed review. We would like to follow up with you regarding our rebuttal.
> > >
> > > Has our response addressed all your questions?
> > >
> > > Please let us know. We are looking forward to hearing back from you.
> > >
> > > Best regards,
> > >
> > > Authors

---

### Author Response · Authors · 2023-11-23
**Rebuttal Summary**

We express our gratitude to all the reviewers for their insightful and constructive feedback. Our contribution is a principled black-box gradient matching algorithm called MATCH-OPT. Our algorithm enables learning effective surrogate models for offline optimization via gradient descent. We theoretically show that our strategy is sound and empirically demonstrate its effectiveness in many practical settings. In response to the reviewer’s comments, we have made a new revision of our paper to incorporate new results and insights from our discussion.

Here, we provide a summary of the key modifications, and where they are incorporated into our revised paper:

**Reviewer 7Vbx:**

We have reported the running time of all baselines (Appendix G)

We have clarified details of the choice of the discretization and other hyper-parameters such as alpha after the description of our loss function (Eq. 12 and Eq. 13).

We have also discussed a potential limitation of our approach in comparison to the baselines and how it was mitigated in the design of our loss function (Appendix H)

**Reviewer ehny:**

We have provided an insight on why organizing training data into monotonically increasing trajectories encourages the model to learn the behavior of a gradient-based optimization algorithm

**Reviewer 1MFJ:**

We have proved that the bound is not loose (see Appendix A, the paragraph after Eq. (22))

We have proved that minimizing Eq. (11) theoretically bound the gradient gap with a quantity that decreases as the loss in Eq. (11) decreases (see the colored paragraph at the end of Appendix A)

We have compared with the suggested baseline DDOM, which shows significant improvement (see Appendix F)

**Reviewer eLSU:**

We have compared with BDI and BO-qEI as requested. The results show that our method performs better and more stable on average (see Appendix F)

We have shown that the assumption that the oracle has gradient has been used in most previous work, including pioneering work in the field such as COMS. We are able to quote exact statements from prior work, such as COMS, stating explicitly that assumption.

We have incorporated a clarification of the terminology “oracle gradient” in our main text (see Section 2), as suggested by reviewer eLSU. This change does not affect the theoretical and practical contribution of our method in any way.

--

We thank all reviewers once again for their detailed feedback.

---

### Meta-Review · Area_Chair_X94Q · 2023-12-14

**Metareview:**

This paper proposes a new algorithm for offline BBO. The key idea is to learn a surrogate to match the gradient field that underlies the data distribution. The authors validate their approach on Design-Bench comparing against a few relevant baselines. While the analysis to better probe the underlying difficulty of the offline BBO problem was interesting, there were several concerns and discussions regarding the algorithmic execution:

* The assumption of an oracle gradient is not practical and quite evidently not a 'black-box' optimization problem. The authors argument that previous works (COMs, BONET, etc.) also makes that assumption is not quite true --- as the reviewer pointed out, it is very different to make an assumption for analysis vs. making it a key part of the algorithm implementation. Moreover, even if there were previous works using the assumption for their experiments (which most, if not all, are not doing), the lack of practical motivation is sufficient ground to discontinue this tradition.

* Related, one other concern was that DDOM is built on diffusion models, which are well-known to have strong connections with score-based models. Yet, they don't assume proxy oracles, so it seems unfair to claim empirical superiority while making a strictly stronger assumption for executing the proposed approach.

Finally, there were some challenges related to the theory where the authors and reviewers disagreed. There was no resolution on that front, so the AC didn't factor that concern in the final decision.

**Justification For Why Not Higher Score:**

Poor motivation and very strong assumptions in the algorithm.

**Justification For Why Not Lower Score:**

N/A

---

### Decision · Program_Chairs · 2024-01-16

Reject